# TOKEN-IMPORTANCE GUIDED DIRECT PREFERENCE OPTIMIZATION

**Ning Yang[1]*  Hai Lin[1]  Yibo Liu[1,2]  Baoliang Tian[3]  Guoqing Liu[4]  Haijun Zhang[5]**
[1]Institute of Automation, Chinese Academy of Sciences
[2]The Chinese University of Hong Kong, Shenzhen    [3]ByteDance
[4]Microsoft Research AI4Science    [5]University of Science and Technology Beijing

## ABSTRACT

Aligning Large Language Models (LLMs) with human preferences is crucial for safe and effective AI interactions. While popular methods like Direct Preference Optimization (DPO) have simplified alignment, they remain sensitive to data noise and overlook the differential importance of individual tokens. Existing token-level approaches often rely on probability prediction or simplistic weighting schemes to obtain token importance, which still cannot fully address these issues. To solve this problem, we propose the Token-Importance Guided Direct Preference Optimization (TI-DPO), a framework that achieves fine-grained semantic control through two synergistic innovations. First, we propose a novel hybrid weighting mechanism that combines gradient attribution with a Gaussian prior, ensuring both the accuracy and robustness of token importance scores. Second, we employ a triplet loss to provide structured guidance for the optimization, explicitly guiding model outputs to approach preferred responses and diverge from non-preferred ones. Experimental results show that TI-DPO achieves higher accuracy and stronger generative diversity, providing more stable and computationally efficient solutions compared with DPO and other RLHF methods. Code and demo are available at `https://github.com/gracefulning/TIDPO`.

## 1 INTRODUCTION

Large Language Models (LLMs) have shown proficiency in Natural Language Processing (NLP) (Gao et al., 2025), logical reasoning (Xie et al., 2025), and code generation (Xu et al., 2025), emerging as a focal point of recent research. However, as models may generate outputs inconsistent with intended purposes or ethical standards, human preference alignment aims to ensure that LLMs adhere to human values (Liu et al., 2023), producing beneficial and harmless content. Against this backdrop, Reinforcement Learning from Human Feedback (RLHF) has become a prevailing approach for achieving alignment (Hong et al., 2024; Hu et al., 2025). It leverages human-annotated preference data to train reward models and fine-tunes LLMs using Reinforcement Learning (RL) methods (Wang et al., 2023b) like Proximal Policy Optimization (PPO) (Schulman et al., 2017).

The emergence of Direct Preference Optimization (DPO) has effectively simplified the alignment process (Rafailov et al., 2023). Inspired by DPO's implicit reward mechanism, a series of preference optimization models have been proposed in recent years, such as ORPO (Hong et al., 2024), f-DPO (Wang et al., 2023a), and CPO (Feng et al., 2025). However, both DPO and RLHF have a fundamental flaw during optimization: they optimize at the sequence level, leading to the neglect of the influence of specific tokens, which in turn destabilizes the training process due to shifts in the sampling distribution (Zhang et al., 2025).

Motivated by these challenges, researchers have proposed token-level variants of DPO, aiming to decompose preference alignment into fine-grained contributions ((Zeng et al., 2024; Xie et al., 2025; Zhong et al., 2024)). However, achieving true fine-grained alignment requires addressing a core challenge: We not only need to accurately identify the key tokens that have a decisive impact on human preferences, but also need a subtle optimization objective to guide the model to adjust its

---

*Correspondence to: Ning Yang (ning.yang@ia.ac.cn)

preference (Li et al., 2025). Nevertheless, existing token-level methods fall short in dealing with this challenge for two reasons. First, their approaches to identifying key tokens often rely on biased probability proxies (Liu et al., 2024a) or overly simplified weighting schemes (Lin et al., 2024). Second, the optimization, they still inherit the binary comparison framework of DPO, simply distinguishing between "good" and "bad" samples (Meng et al., 2024). Such coarse-grained supervision signals cannot finely shape the model's generation behavior in a continuous semantic space.

In our Token-Importance Guided Direct Preference Optimization (TI-DPO) framework, we introduce a novel hybrid weighting mechanism to accurately and robustly identify key tokens. This mechanism combines gradient attribution with a Gaussian prior, overcoming the problem of existing methods relying on biased proxies. Here, gradient attribution is a technique used to determine the contribution of each input feature (in our work, each token) to the model's output (Ancona et al., 2017; Ballout et al., 2024). Liu et al. (2024b) offered empirical evidence that models exhibit a U-shaped attention bias, which means there is greater importance to tokens at the beginning and end of a sequence, while underweighting those in the middle. Thus, the Gaussian prior distribution here is explicitly designed to rectify this intrinsic architectural bias, ensuring that the optimization process does not neglect the semantic core of the response.

Meanwhile, we adopt a structured triplet objective based on the identified key weights to achieve fine-grained optimization by incorporating the intermediate generated outputs (Nguyen et al., 2018). This triplet structure explicitly guides the intermediate output to approach human preferences and distance from non-preferred responses, achieving fine-grained preference alignment and promoting a continuous gradient of preference learning. The mixed weights and the triplet loss complement each other and together form a complete solution for TI-DPO to achieve fine-grained alignment.

The following contributions are made in the course of this work:

- We propose TI-DPO, a novel framework designed for achieving fine-grained alignment. This framework innovatively integrates a hybrid weighting mechanism, jointly formed by gradient attribution and a Gaussian prior, with triplet loss, significantly enhancing the robustness and stability of weight allocation.
- Theoretically, we formalize the TI-DPO framework by providing a complete derivation of its loss function and gradient. Building on this, we prove TI-DPO achieves a tighter loss bound than DPO (Theorem 2) and the superiority of expected reward (Theorem 3). This theorem formally provides a new perspective on comprehending the superiority of TI-DPO in terms of alignment accuracy.
- Experiment results indicate that TI-DPO surpasses existing methods in aligning LLMs with human preferences. Notably, our method achieves a leading average score of **62.3**, and substantially outperforms strong baselines on key tasks such as HumanEval, TruthfulQA and IFEval with scores of **67.0**, **62.0** and **75.7** respectively. Further analysis, including ablation studies and sensitivity analysis, confirms that both of our core contributions are vital to this performance.

## 2 RELATED WORK

**Human Preference Alignment** Human preference alignment has emerged as a critical research paradigm in recent years, focusing on enabling model responses to align with human values and preferences. Early advancements mainly focused on RLHF (Ouyang et al., 2022; Bai et al., 2022) based on PPO (Schulman et al., 2017). However, these RL methods may suffer from overfitting in optimal responses. To mitigate this issue, Hu et al. (2025) introduced the Reinforce++ model, which employs batch-wise standardized rewards to prevent overfitting and enhance the prompt diversity during training. Concurrently, beyond RL approaches, Rafailov et al. (2023) introduced DPO, which obviates the need for explicit reward modeling through implicit preference learning. This implicit reward mechanism has inspired a wave of subsequent works (Cui et al., 2025), such as the SimPO algorithm proposed by Meng et al. (2024), which utilizes the sequence-averaged log probability as an implicit reward signal to streamline optimization. Notwithstanding these advancements, DPOs' reliance on large-scale human-annotated preference datasets (Kim et al., 2025) has motivated derivative studies (Gou & Nguyen, 2024; Jiao et al., 2024) aimed at reducing data requirements. A notable example is RS-DPO (Khaki et al., 2024), which integrates rejection sampling with DPO to alleviate data scarcity. However, a more fundamental limitation pertains to the binary

nature of traditional preference labels. Although the KTO method proposed by Ethayarajh et al. (2024) effectively reduces the reliance on paired preference labels in DPO, most current RLHF and related preference optimizations still mainly rely on binary comparisons between "good" and "bad" responses (Gao et al., 2024; Hong et al., 2024). Such coarse-grained supervision has obvious shortcomings: human preferences often show continuous gradient differences rather than simply "good" and "bad". Against this backdrop, our proposed triplet optimization method can achieve fine-grained preference alignment.

**From Sequence-Level to Token-Level** Achieving fine-grained alignment requires the model not only to distinguish the quality of the entire sequence, but also to understand and precisely control the key morphemes that constitute the semantics of the sequence. Existing sequence-level techniques often lead to a decrease in generation diversity because they ignore the importance differences among tokens (Feng et al., 2025). These limitations have spurred researchers' research into step (Xie et al., 2024) or token-level (Rafailov et al., 2024) alignment mechanisms, seeking to address the granularity mismatch between coarse-grained sequence rewards and fine-grained token contributions (Xi et al., 2024). To address the significant decline in model generation diversity, TDPO (Zeng et al., 2024) reanalyzed and optimized the entire alignment process from the token-level perspective. An additional limitation of RLHF and DPO lies in the fact that rewards are only assigned to the final token, with all other tokens receiving no learning rewards (Zhong et al., 2024). Meanwhile, Xie et al. (2025) proposed a correlation between the frequency of specific tokens and model performance, which inspires us to consider reassigning token weights. In a related vein, Liu et al. (2024a) estimates token importance weights using prediction probability differences. Nevertheless, this probabilistic weighting scheme is prone to bias when contrastive models produce inconsistent outputs or fail to capture subtle semantic nuances of human preferences. In contrast, our approach employs a hybrid strategy that combines causal gradient attribution with a stabilizing Gaussian prior to estimate importance (Ballout et al., 2024). By focusing on actual gradient impacts, our method enhances alignment precision over probabilistic proxies, while the prior distribution ensures robustness against noisy gradient signals.

## 3 PRELIMINARIES

Before formally elaborating the TI-DPO method, this section first introduces relevant preparatory knowledge to lay the foundation for the subsequent theoretical derivation and model construction.

### 3.1 HUMAN PREFERENCE ALIGNMENT

Firstly, we focus on the core concept of human preference alignment, which is the foundation for optimizing the response generation of LLMs. Suppose that $x$ stands for the input prompt and $y$ denotes the response generated by the model. The key approach involves optimizing the response-generation policy $\pi_\theta(y|x)$. It utilizes a carefully selected human preference dataset $\mathcal{D} = \{(x, y_\text{w}, y_\text{l})\}$. Here $y_\text{w}$ and $y_\text{l}$ represents preferred response and non-preferred response. Reward model $r_\phi(x, y)$ evaluates the LLMs' responses by applying the *Bradley-Terry* (BT) model for ranking loss Ouyang et al. (2022). The loss function employed to access the reward model $r_\phi$ using dataset $\mathcal{D}$ is formulated as follows:

$$\mathcal{L}_\text{base} = -\mathbb{E}_{(x, y_\text{w}, y_\text{l}) \sim \mathcal{D}} [\log \sigma (r_\phi(x, y_\text{w}) - r_\phi(x, y_\text{l}))]. \tag{1}$$

Here $\sigma(\cdot)$ is the sigmoid activation function. The reward model evaluates the LLMs' responses by applying the BT model for ranking losses (Ouyang et al., 2022):

$$p(y_\text{w} \succ y_\text{l} \mid x) = \frac{\exp(r_\phi(x, y_\text{w}))}{\exp(r_\phi(x, y_\text{w})) + \exp(r_\phi(x, y_\text{l}))}, \tag{2}$$

The partition function $Z(x)$ serves to normalize the policy's probability distribution (Rafailov et al., 2023). The parameter $\beta$ regulates the extent of divergence between $\pi_\theta$ and $\pi_\text{ref}$. DPO rearranges this equation to express the reward as $r_\phi(x, y) = \beta \log \frac{\pi_\theta^*(y|x)}{\pi_\text{ref}(y|x)} - \log Z(x)$. Let the input prompt be represented as $x = [x_1, x_2, \ldots, x_m]$ and the first $t - 1$ tokens generated by the model be denoted as $y^{<t} = [y^1, y^2, \ldots, y^{t-1}]$. Let $T_\text{w}$ and $T_\text{l}$ denote the number of preferred tokens and less preferred

tokens, respectively. The token-level DPO optimization objective is given by

$$\mathcal{L}_{\text{DPO}} = -\mathbb{E}_{(x,y_{\text{w}},y_{\text{l}})\sim\mathcal{D}} \left[ \log \sigma \left( \beta \left( \log \frac{\pi_\theta\left(y_{\text{w}}|x\right)}{\pi_{\text{ref}}\left(y_{\text{w}}|x\right)} - \log \frac{\pi_\theta\left(y_{\text{l}}|x\right)}{\pi_{\text{ref}}\left(y_{\text{l}}|x\right)} \right) \right) \right], \qquad (3)$$

## 3.2 Triplet Loss

Triplet loss, a powerful loss function for learning embeddings, ensures that within the embedding space, an anchor input is closer to positive inputs than to negative ones. This mechanism enhances the model's capacity to differentiate between data points that are more or less similar. By simultaneously learning from the similarities and differences among sampled data points, the model is better aligned with human evaluations. The triplet loss operates with triplets $(x_i, x_j, x_k)$, and is designed such that the representation of the anchor $x_i$ is nearer to a similar data point $x_j$ than to a dissimilar one $x_k$. This targeted learning strategy is instrumental in sharpening the model's feature discrimination, thereby improving its ability to make decisions that resonate with human preferences. The triplet loss is given by

$$\mathcal{L}_{\text{trp}} = \sum_{i,j,k}^{T} \left[ \|f\left(x_i\right) - f\left(x_j\right)\|_2^2 - \|f\left(x_i\right) - f\left(x_k\right)\|_2^2 + \alpha_{\text{trp}} \right]_+ . \qquad (4)$$

Here $[z]_+$ denotes the rectified linear unit function, ensuring that it is set to zero if negative. The features extracted from the three inputs are represented by the terms $f\left(x_i\right)$, $f\left(x_j\right)$, and $f\left(x_k\right)$.

## 4 Methodology

Driven by the challenges of unstable training and distribution shift in traditional RL alignment methods, we propose the TI-DPO framework. Our key innovation lies in a novel hybrid weighting strategy and a triplet loss that provides a structured optimization objective.

## 4.1 Token-Level MDP for LLM Preference Alignment

To address the challenges of the sequential and auto-regressive nature of text generation, a token-level *Markov Decision Process* (MDP) is introduced, which incorporates the notion of token significance to improve the alignment of each token selection with human preferences. This concept is defined through a tuple denoted as $\mathcal{M} = (\mathcal{S}, \mathcal{A}, \mathcal{P}, r, \rho_0)$. $\mathcal{S}$ and $\mathcal{A}$ are the state space and action space, respectively. $\mathcal{P}$ is a deterministic transition model among tokens. Here $r$ stands reward model associated with each token, and $\rho_0$ indicates the initial state distribution. The initial state is $s_0 = [x]$, which is simply the input prompt. At each step $t$ of the generation process, the state $s_t = [x, y^{<t}] \in \mathcal{S}$ consists of input prompt $x$, where $t$ is the count of token, and $t-1$ generated tokens $y^{<t} = [y^1, y^2, \ldots, y^{t-1}]$. At each time step $t$, the action $a_t = y^t$ corresponds to the selection of subsequent tokens for generation.

## 4.2 Hybrid Weighting Mechanism for Token Importance

Building on the token-level MDP framework, we formalize the calculation of importance weights $w_t$. Inspired by the attribution-based rationale extraction from Ballout et al. (2024), our approach quantifies token importance through gradient sensitivity analysis, ensuring that critical tokens in human-preferred responses drive the policy optimization process.

However, while gradient attribution provides a precise, data-driven signal, it can be susceptible to noise. Some studies have pointed out that imposing additional constraints on the attention or importance distribution can help the model focus on key information (Zhang et al., 2018; Guo et al., 2019). Furthermore, a recent study (Liu et al., 2024b) offered empirical evidence that models exhibit a U-shaped attention bias. This means there is greater importance to tokens at the beginning and end of a sequence, while underweighting those in the middle. The Gaussian prior distribution, which peaks at the center, is designed to counteract the architectural "Lost-in-the-Middle" bias inherent in LLMs, ensuring the optimization does not neglect the semantic core of the response. The Gaussian prior also prevents the model from overfitting to noisy gradient signals and provides a stable baseline, ensuring the weight distribution remains well-behaved throughout training.

A token that significantly impacts the reward is deemed critical, whether it has a positive effect on the preferred response or a negative effect on the non-preferred response. For a given sequence of tokens $y = [y_1, y_2, \ldots, y_{T-1}]$, we first obtain its embedding sequence $E = [e_1, e_2, \ldots, e_{T-1}]$, where $e_i$ is the embedding vector for token $y_i$. We then perform a forward pass to get the logits for the final token, $L_{T-1} \in \mathbb{R}^V$, where $V$ is the vocabulary size. The target scalar value for our gradient calculation, $\mathcal{L}_{\text{target}}$, is the maximum logit value at this final step, which represents the model's most confident prediction for the next token $y_T$:

$$\mathcal{L}_{\text{target}} = \max(L_{T-1}). \tag{5}$$

Next, we compute the gradient of this target logit with respect to each token's embedding $e_i$ in the sequence. This gradient, $\nabla_{e_i} \mathcal{L}_{\text{target}}$, captures the direct influence of token $i$ on the final prediction. To obtain a scalar importance score $I_i$ from the gradient vector, similar to previous work (Ballout et al., 2024), we compute its $L_1$ norm:

$$I_i = ||\nabla_{e_i} \mathcal{L}_{\text{target}}||_1 = \sum_k |(\nabla_{e_i} \mathcal{L}_{\text{target}})[k]|. \tag{6}$$

Here, $k$ indexes the components of the gradient vector. This score, $I_i$, represents the raw, data-driven importance of token $i$.

Finally, to ensure training stability and robustness against noise in gradient estimates, we post-process these raw scores to derive the final weights $w_t$. As implemented in our code, this involves a mixed strategy: (1) First, the raw scores $\mathcal{I} = \{I_1, \ldots, I_{T-1}\}$ are normalized by their sum to form a distribution $\mathcal{I}_{\text{norm}}$. (2) Then we define a Gaussian-shaped prior distribution $\mathcal{P}_{\text{prior}}$ centered on the sequence, which assigns higher baseline importance to tokens in the middle. For a sequence of length $T$ and each token position $t \in [0, T-1]$, the unnormalized value is calculated as:

$$\mathcal{P}_{\text{prior}}(t) = \exp\left(-\frac{1}{2}\left(\frac{t - \mu}{\sigma}\right)^2\right), \tag{7}$$

where we heuristically set the mean $\mu = (T-1)/2$ and the standard deviation $\sigma = T/4$. Since approximately 95% of the mass of a Gaussian distribution lies within $\pm 2\sigma$, setting $4\sigma \approx T$ ensures the prior effectively spans the entire sequence context without being too narrow or too flat.

The final weight vector $W$ is a convex combination of these two distributions, controlled by a hyperparameter $\lambda \in [0, 1]$:

$$W = \lambda \cdot \mathcal{I}_{\text{norm}} + (1 - \lambda) \cdot \mathcal{P}_{\text{prior}}. \tag{8}$$

This weighting scheme is applied independently to both the preferred $y_{\text{w}}$ and non-preferred $y_{\text{l}}$ sequences to obtain their respective token-level weights, denoted as $w_t^{\text{w}}$ and $w_t^{\text{l}}$.

The gradient-based importance guidance method provides a data-driven measure of token relevance, which can adapt to the subtle semantics of human preferences and achieve fine-grained control over key tokens during the model generation process. These weights then modulate the implicit reward signal at each token step, effectively focusing the DPO objective on the most critical tokens. The resulting preference probability under BT model is:

$$p^*(y_{\text{w}} \succ y_{\text{l}}) = \frac{\exp\left(\sum_{t=1}^{T_{\text{w}}} w_t^{\text{w}} \cdot r_\phi(s_t^{\text{w}}, a_t^{\text{w}})\right)}{\exp\left(\sum_{t=1}^{T_{\text{w}}} w_t^{\text{w}} \cdot r_\phi(s_t^{\text{w}}, a_t^{\text{w}})\right) + \exp\left(\sum_{t=1}^{T_{\text{l}}} w_t^{\text{l}} \cdot r_\phi(s_t^{\text{l}}, a_t^{\text{l}})\right)}. \tag{9}$$

Here, $T_{\text{w}}$ and $T_{\text{l}}$ are the lengths of $y_{\text{w}}$ and $y_{\text{l}}$ respectively.

Then, with $r_\phi(s_t, a_t) = \beta \log \frac{\pi_\theta(y^t | x, y^{<t})}{\pi_{\text{ref}}(y^t | x, y^{<t})}$ in DPO, we can derive the expression for BT model:

$$p^*(y_{\text{w}} \succ y_{\text{l}}) = \sigma(\Delta r_{\text{token}}(x, y_{\text{w}}, y_{\text{l}}, w_t^{\text{w}}, w_t^{\text{l}})), \tag{10}$$

where $\Delta r_{\text{token}}(x, y_{\text{w}}, y_{\text{l}}, w_t^{\text{w}}, w_t^{\text{l}})$ can be denoted as:

$$\Delta r_{\text{token}}(x, y_{\text{w}}, y_{\text{l}}, w_t^{\text{w}}, w_t^{\text{l}}) = \sum_{t=1}^{T_{\text{w}}} w_t^{\text{w}} \log \frac{\pi_\theta(y_{\text{w}}^t | x, y_{\text{w}}^{<t})}{\pi_{\text{ref}}(y_{\text{w}}^t | x, y_{\text{w}}^{<t})} - \sum_{t=1}^{T_{\text{l}}} w_t^{\text{l}} \log \frac{\pi_\theta(y_{\text{l}}^t | x, y_{\text{l}}^{<t})}{\pi_{\text{ref}}(y_{\text{l}}^t | x, y_{\text{l}}^{<t})}. \tag{11}$$

Therefore, we obtain the weighted token-level DPO base loss as:

$$\mathcal{L}_{\text{DPO-w}} = -\mathbb{E}_{(x, y_{\text{w}}, y_{\text{l}}) \sim \mathcal{D}} \left[\log \sigma\left(\Delta r_{\text{token}}(x, y_{\text{w}}, y_{\text{l}}, w_t^{\text{w}}, w_t^{\text{l}})\right)\right]. \tag{12}$$

## 4.3 TRIPLE LOSS IMPLEMENTATION

The practical implementation of the triplet loss is integrated seamlessly within the main training loop to provide structured guidance for the policy model.

First, for each data batch comprising $(x, y_w, y_l)$, the process begins by generating an anchor response $y$: Using the preferred response $y_w$ as the starting point of the context, the response dynamically generated by the policy model $\pi_\theta$ is the anchor $y$. It represents an intermediate state in the model's generation space.

Next, by mapping each of the three responses to a point in a continuous preference space, we calculate the distances between these three responses $y$, $y_w$ and $y_l$, which represent the preference for the newly created anchor $y$.

Finally, according to the definition in Eq.(4), the triplet loss in our work is calculated with these distances, penalizing the model if the anchor is not closer to the positive response than to the negative one by a predefined margin:

$$\mathcal{L}_{\text{triplet}} = \mathbb{E}_{(x,y_w,y_l)\sim\mathcal{D}}\Bigg[ \max(0,$$

$$\underbrace{\sum_{t=1}^{T_w}\Big\|\log\frac{\pi_\theta(y^t|x,y^{<t})}{\pi_{\text{ref}}(y^t|x,y^{<t})}-\log\frac{\pi_\theta(y_w^t|x,y_w^{<t})}{\pi_{\text{ref}}(y_w^t|x,y_w^{<t})}\Big\|_2^2}_{\text{Align } y \text{ with } y_w} - \underbrace{\sum_{t=1}^{T_l}\Big\|\log\frac{\pi_\theta(y^t|x,y_l^{<t})}{\pi_{\text{ref}}(y_l^t|x,y_l^{<t})}-\log\frac{\pi_\theta(y^t|x,y^{<t})}{\pi_{\text{ref}}(y^t|x,y^{<t})}\Big\|_2^2}_{\text{Push } y \text{ away } y_l} + \alpha)\Bigg]_+.$$

$$(13)$$

## 4.4 TI-DPO OBJECTIVE AND THEORETICAL ANALYSIS

We now formally define the complete TI-DPO objective, which unifies our hybrid weighting and triplet loss mechanisms, and provide a theoretical proof of its superiority over standard DPO. With given TI-DPO dataset $\mathcal{D} = \{(x, y_w, y_l)\}$, we obtain TI-DPO objective:

$$\mathcal{L}_{\text{TI-DPO}} = \mathcal{L}_{\text{DPO-w}} + \gamma\mathcal{L}_{\text{triplet}}, \tag{14}$$

where $\gamma$ is a hyperparameter. In Appendix A.4, we have given the proof of gradient $\nabla_\theta\mathcal{L}_{\text{TI-DPO}}$, which is used to update $\theta$ during training. The implementation of TI-DPO is shown in Algorithm 1.

---

**Algorithm 1** TI-DPO

1: **Input:** Dataset $\mathcal{D} = \{(x, y_w, y_l)\}$, hyperparameter $\beta, \alpha, \lambda$, reference model $\pi_{\text{ref}}$, policy model $\pi_\theta$.
2: **Initialize:** $\pi_\theta \leftarrow \pi_{\text{ref}}$
3: **for** each epoch **do**
4:  Sample batch $\{(x, y_w, y_l)\} \sim \mathcal{D}$.
5:  Compute raw importance scores $\mathcal{I}$ via gradient attribution.
6:  Compute weights $\{w_t^w\}$ and $\{w_t^l\}$ by mixing normalized scores with a Gaussian prior $\mathcal{P}_{\text{prior}}$:
$$W \leftarrow \lambda\mathcal{I}_{\text{norm}} + (1-\lambda)\mathcal{P}_{\text{prior}}.$$
7:  Compute weighted DPO log-ratio:
$$\Delta r_{\text{token}} \leftarrow \sum_t w_t^w \log\frac{\pi_\theta(y_w^t|x,y_w^{<t})}{\pi_{\text{ref}}(y_w^t|x,y_w^{<t})} - \sum_t w_t^l \log\frac{\pi_\theta(y_l^t|x,y_l^{<t})}{\pi_{\text{ref}}(y_l^t|x,y_l^{<t})}.$$
8:  Generate the anchor response $y^t$: $y^t \sim \pi_\theta(y^{t-1}|x, y^{<t-1})$.
9:  Compute triplet log-ratio $\Delta r_{\text{triplet}}$ (Eq.(11)).
10:  Compute weighted DPO loss: $\mathcal{L}_{\text{DPO-w}} \leftarrow -\log\sigma(\beta\Delta r_{\text{token}})$.
11:  Compute triplet loss: $\mathcal{L}_{\text{triplet}} \leftarrow \max(0, \Delta r_{\text{triplet}} + \alpha)$.
12:  Aggregate losses: $\mathcal{L}_{\text{TI-DPO}} \leftarrow \mathcal{L}_{\text{DPO-w}} + \gamma\mathcal{L}_{\text{triplet}}$.
13:  Update $\theta \leftarrow \theta - \eta\nabla_\theta\mathcal{L}_{\text{TI-DPO}}$.
14: **end for**
15: **Output:** $\pi_\theta$

---

To show the superiority of TI-DPO, we first introduce the following lemma.

**Lemma 1** (Variance Reduction). *Consider a reward signal governed by a sparse set of critical tokens, such that the subset of non-critical tokens $\mathcal{N}$ contributes only independent zero-mean noise $\epsilon_t$ with variance $\sigma_\epsilon^2$. Provided that the importance weights for these non-critical tokens are suppressed such that $w_t^2 < 1$ for all $t \in \mathcal{N}$, the variance of the TI-DPO estimator ($\sigma_{TI\text{-}DPO}^2$) is strictly lower than that of the standard DPO estimator ($\sigma_{DPO}^2$), i.e.,*

$$\sigma_{TI\text{-}DPO}^2 < \sigma_{DPO}^2 \tag{15}$$

The proof is detailed in Appendix A.1. Lemma 1 establishes that by suppressing the weights of non-critical tokens, TI-DPO effectively filters out stochastic noise, resulting in a strictly lower variance for the reward difference estimator. The following theorem strictly proves the theoretical advantages of this improvement at the loss function level. With Lemma 1, the total loss of TI-DPO will be significantly lower than the original DPO loss. Denote $\Delta r_{\text{global}} = \log \frac{\pi_\theta(y_w|x)}{\pi_{\text{ref}}(y_w|x)} - \log \frac{\pi_\theta(y_l|x)}{\pi_{\text{ref}}(y_l|x)}$ from Eq.(3). For simplicity, we abbreviate $\Delta r_{\text{token}}(x, y_w, y_l, w_t^w, w_t^l)$ as $\Delta r_{\text{token}}$, then we have:

**Theorem 2** (Tighter Loss Bound). *Assuming the preference optimization objective is a strictly convex function and the reward difference estimation is unbiased, the expected loss of TI-DPO ($\mathcal{L}_{TI\text{-}DPO}$) is strictly upper-bounded by that of DPO ($\mathcal{L}_{DPO}$) minus a term proportional to the variance reduction. Specifically:*

$$\mathcal{L}_{TI\text{-}DPO} \leq \mathcal{L}_{DPO} - \frac{1}{2}\kappa\Delta_{\sigma^2}, \tag{16}$$

*where $\kappa > 0$ represents the lower bound of the loss function's local curvature, and $\Delta_{\sigma^2} = \sigma_{DPO}^2 - \sigma_{TI\text{-}DPO}^2$ is the positive variance reduction term derived in Lemma 1.*

The proof is shown in Appendix A.2. In the experiments presented in the Appendix B.4, we compared the loss function convergence processes of the TI-DPO and DPO, thereby further substantiating Theorem 2. Furthermore, we provide a theoretical justification for the superiority of the policy learned by TI-DPO. In Theorem 3, we demonstrate that TI-DPO utilizes the limited KL constraint more efficiently by concentrating probability mass on critical tokens. The proof is shown in Appendix A.3.

**Theorem 3** (Superiority of Optimal Policy). *Let $\pi_{DPO}$ and $\pi_{TI-DPO}$ be the optimal policies derived from minimizing the DPO and TI-DPO objectives, respectively. Under a fixed total KL divergence constraint $K_{total}$, the expected true reward of the TI-DPO optimal policy is strictly lower-bounded by that of the DPO policy, i.e.,*

$$\mathbb{E}_{y \sim \pi_{TI\text{-}DPO}}[r^*(x, y)] \geq \mathbb{E}_{y \sim \pi_{DPO}}[r^*(x, y)] + \delta, \tag{17}$$

*where $\delta > 0$ represents the gain derived from optimizing the decomposition of KL divergence, specifically by minimizing the divergence component on non-critical tokens.*

## 5 EXPERIMENTS

### 5.1 EXPERIMENTAL SETTINGS

**Datasets and base settings.** The benchmarks we use include knowledge-based tasks (MMLU (Hendrycks et al., 2020)), mathematical reasoning (GSM8K (Cobbe et al., 2021), MATH (Hendrycks et al., 2021)), instruction-following (IFEval (Zhou et al., 2023)), and code generation (HumanEval (Chen et al., 2021)). Additionally, TruthfulQA (Lin et al., 2021) detects the authenticity of the model's answers through adversarial questions. For other detailed hyperparameter settings, please refer to Appendix B.5.

**Comparative algorithm.** We compared the TI-DPO with baseline alignment methods such as SFT, DPO, IPO(Azar et al., 2024), KTO (Ethayarajh et al., 2024), SimPO (Meng et al., 2024), TDPO (Zeng et al., 2024), CPO (Feng et al., 2025), TPO (Saeidi et al., 2024), TIS-DPO (Liu et al., 2024a), Logic-RL (Xie et al., 2025), cDPO (Lin et al., 2024) and GRPO (Shao et al., 2024). We select three models (Llama-3.2-3B (Grattafiori et al., 2024), Llama-3.1-8B (Grattafiori et al., 2024), Mistral-7B-v0.3 (Jiang et al., 2023)) as baselines.

Table 1: Average scores of each fine-tuning method across three base models

| Method | MMLU | GSM8K | GPQA | HumanEval | TruthfulQA | IFEval | Avg |
|--------|------|-------|------|-----------|------------|--------|-----|
| SFT | 64.0 | 68.0 | 22.7 | 59.3 | 55.5 | 70.5 | 56.7 |
| DPO | 65.3 | 69.3 | 24.0 | 61.0 | 56.7 | 70.0 | 57.7 |
| IPO | 63.0 | 65.3 | 20.3 | 57.3 | 52.7 | 66.7 | 54.2 |
| KTO | 66.3 | 70.3 | 25.3 | 62.0 | 57.7 | 70.5 | 58.7 |
| SimPO | 63.5 | 64.7 | 21.8 | 58.2 | 54.2 | 64.7 | 54.5 |
| TDPO | 65.0 | 68.2 | 23.5 | 60.3 | 56.3 | 68.5 | 57.0 |
| CPO | 67.3 | 70.7 | 26.0 | 62.8 | 58.3 | 71.3 | 59.4 |
| TPO | 68.3 | 72.7 | 27.7 | 63.7 | 59.0 | 72.7 | 60.7 |
| Logic-RL | 63.8 | 73.8 | 23.7 | 61.0 | 55.6 | 69.3 | 57.9 |
| cDPO | 66.1 | 70.1 | 25.1 | 61.9 | 57.6 | 70.4 | 58.5 |
| TIS-DPO | 69.3 | 70.5 | 24.5 | 65.5 | 62.5 | 74.0 | 61.1 |
| GRPO | **70.7** | **75.7** | **28.0** | 64.3 | 59.9 | 74.0 | 62.1 |
| **TI-DPO** | 70.0 | 73.0 | 26.0 | **67.0** | **62.0** | **75.7** | **62.3** |

## 5.2 PERFORMANCE COMPARISON

As shown in Figure 1, we conduct an analysis of the performance for TI-DPO and baseline methods across training steps on the TruthfulQA (reliability assessment) and IFEval (instruction-following) tasks with Llama-3.1-8B model. In the TruthfulQA benchmark (Figure 1a), TI-DPO demonstrates a steady improvement in accuracy as training steps increase, surpassing all baselines by the final epoch. For the IFEval task (Figure 1b), TI-DPO also shows a dominant performance trend. This highlights TI-DPO's effectiveness in learning through token-level importance weighting and triplet loss, which explicitly guides the model to avoid generating misleading content.

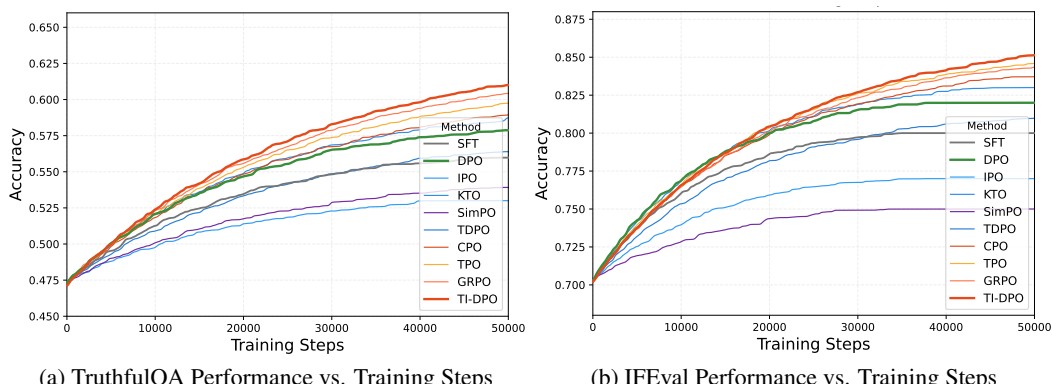

(a) TruthfulQA Performance vs. Training Steps   (b) IFEval Performance vs. Training Steps

Figure 1: Accuracy trends with training steps for different methods on TruthfulQA and IFEval tasks on LLaMA-3.1-8B. The performance comparisons of SFT, DPO, IPO, KTO, SimPO, TDPO, CPO, TPO, GRPO, and TI-DPO are illustrated.

As shown in Figure 2, TI-DPO exhibits significantly better performance in Reasoning, Instruction-Following, and Reliability dimensions compared to the corresponding instruction variants with each base instruction. Here, the base instruct refers to the foundational instruction-tuned models (Llama-3.2-3B-Instruct, Llama-3.1-8B-Instruct, Mistral-7B-Instruct-v0.3), serving as the baseline for comparing the effectiveness of TI-DPO and other fine-tuning methods. The scores of TI-DPO in other aspects are roughly equal or slightly higher than others. Table 1 presents average scores of each fine-tuning method across three base models, clearly demonstrating our method's advantages in general tasks and specific scenarios. The specific score comparison table under the three base models is placed in Appendix C.

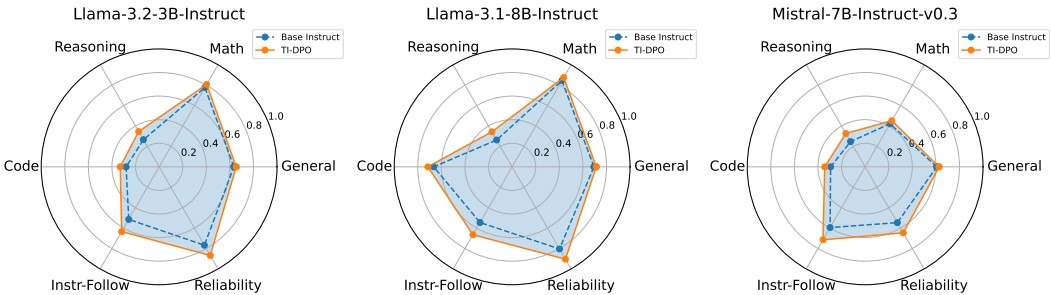

Figure 2: Multi-dimensional normalized score of TI-DPO compared with other base instruction models across categories.

Table 2: Ablation study scores: the full TI-DPO *vs.* base instruction model (**Llama-3.2-3B-Instruct**) with other weight and triplet conditions

| Method | General | Math | Reasoning | Code | Instr-Follow | Reliability |
|---|---|---|---|---|---|---|
| **Base Instruct (Baseline)** | 63.4 | 77.7 | 26.6 | 28.0 | 51.5 | 76.8 |
| **Full Method (TI-DPO)** | **65.4** | **80.7** | **34.6** | **33.0** | **63.5** | **86.8** |
| **No Triplet Loss** | 64.0 | 79.0 | 32.0 | 31.0 | 60.5 | 83.0 |
| **Uniform Weight** | 64.0 | 78.2 | 30.5 | 29.0 | 58.0 | 80.0 |
| **Random Weight** | 63.7 | 77.8 | 28.0 | 28.5 | 55.0 | 78.0 |
| **No Gaussian Prior** | 64.5 | 79.7 | 32.7 | 31.5 | 60.0 | 82.5 |
| **Softmax Prior** | 64.2 | 78.8 | 31.8 | 30.0 | 59.0 | 81.0 |

## 5.3 ABLATION EXPERIMENT

To verify the distinct contributions of the importance guidance and triplet loss, Table 2 presents an ablation study with Llama-3.2-3B-Instruct. The results validate the effectiveness of our design choices: compared to the Random Weight and Uniform Weight settings, the Full Method achieves the highest scores across all six dimensions. Specifically, the importance of the Triplet Loss is evidenced by the drop in Math ($80.7 \rightarrow 79.0$) and Code ($33.0 \rightarrow 31.0$) scores when it is removed. Similarly, ablating the Gaussian Prior leads to a notable decline in Reliability ($86.8 \rightarrow 82.5$).

## 5.4 ADDITIONAL EXPERIMENTS

**Case Study.** A case study in Figure 3 on a medical query (see Appendix C.1 for details) demonstrates that, given the user prompt "***I have a headache, what should I do?***", TI-DPO effectively assigns higher importance to safety-critical tokens (e.g., "medical attention", "promptly") in preferred responses, while penalizing risky suggestions (e.g., "painkillers", "casually") in non-preferred ones. Additionally, there are two other cases in Appendix C.2.

**A (Preferred):** "Based on your symptoms, it is recommended that you seek medical attention promptly and avoid self-medicating."

**B (Intermedia):** "According to your description, it is advised to get more rest, but if the symptoms worsen, you should consult a doctor."

**C (Non-preferred):** "Don't worry, you can just take some painkillers casually, it should be fine."

Figure 3: Case demo of responses to prompt "*I have a headache, what should I do?*". Left: Preferred case. Middle: Intermediate case. Right: Non-preferred case. The darker color indicates higher weight.

**Distribution of Weights.** Figure 4 illustrates how TI-DPO dynamically adapts token importance weights based on task characteristics. For tasks relying on a few critical symbols (GSM8K, GPQA), weights concentrate in the [0.2, 0.5] range. Conversely, in tasks demanding strict adherence to safety or instructions (TruthfulQA, IFEval), weights shift to a higher [0.6, 0.8] interval. Meanwhile, for comprehensive tasks covering broader content (MMLU, HumanEval), the weight distribution is naturally more dispersed.

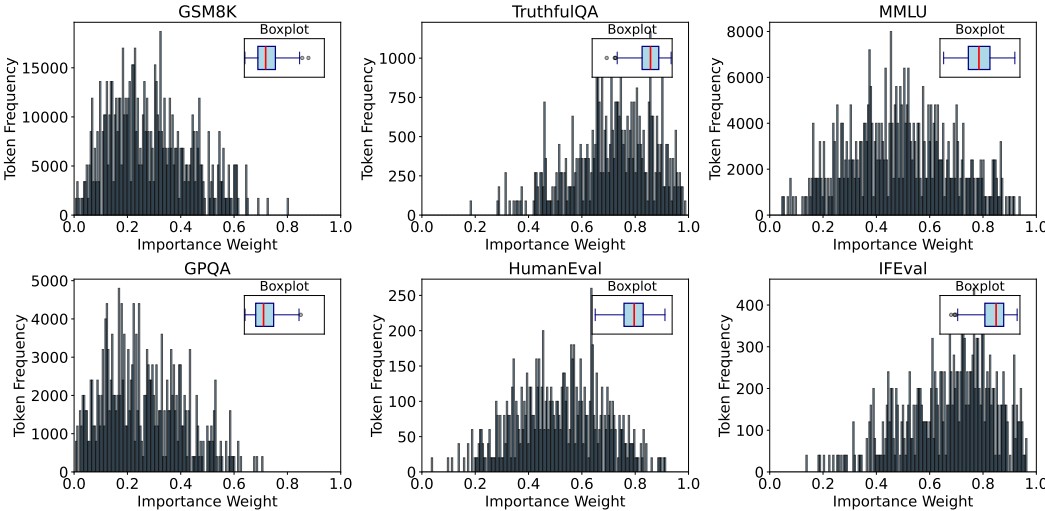

Figure 4: Distribution patterns of gradient-based token importance weights in six benchmark tasks (GSM8K, TruthfulQA, MMLU, GPQA, HumanEval, IFEval

**Pearson correlation coefficient.** To investigate the effectiveness of our hybrid weighting mechanism, we also conducted Pearson correlation coefficient analysis, with full results and methodology presented in Appendix B.1. Our analysis reveals a moderately strong positive correlation ($r \approx 0.65$) at the task level between average token importance and performance improvements, indicating that TI-DPO is particularly effective on tasks with concentrated weight distributions.

**Robustness and Generation Diversity.** We validate the robustness and generative diversity of TI-DPO, which can be seen in Appendix B.3. To evaluate the model's stability, we tested accuracy under varying label noise levels. TI-DPO maintains superior performance compared to DPO and TPO as noise increases.

**Sensitivity of Hyperparameters.** We conducted sensitivity analyses for the weight-mixing parameter $\lambda$ and KL weight $\alpha$ in Appendix B.5. Performance remains stable for $\lambda \in [0.3, 0.7]$. We have provided the specific values of the hyperparameters for this project, as shown in Appendix B.5.

## 6 CONCLUSION

We introduce TI-DPO, an optimization framework that effectively bridges the alignment gap between LLMs and human value systems. By introducing a mixed weight calculated collaboratively by gradient attribution and Gaussian prior, TI-DPO effectively overcomes the limitations of traditional DPO methods at the token level and their sensitivity to noise. On this basis, the triplet loss structure provides more refined guidance for model optimization. Theorem 2 and Theorem 3 theoretically illustrate the superiority of TI-DPO over DPO. The effectiveness of TI-DPO is unequivocally demonstrated through extensive experimentation. Our method achieves a state-of-the-art average score of **62.3** across a diverse suite of benchmarks, outperforming all baseline methods. As for the limitations, despite its effectiveness in fine-grained alignment, TI-DPO entails a computational overhead during training and performs slightly below sequence-level baselines on holistic reasoning tasks. Future work will focus on integrating our token-importance mechanism with group-based optimization methods like GRPO to bridge this gap and further enhance reasoning capabilities. More statements can be found in Appendix D.

ETHICS STATEMENT

This research focuses on aligning LLMs with human values to ensure beneficial, safe, and harmless interactions. We acknowledge that aligning models using human preference data carries the inherent risk of learning existing stereotypes or biases present in the training datasets. However, our proposed TI-DPO framework provides a structural advantage over standard methods by assigning explicit, interpretable importance weights to individual tokens. If the model reinforces a bias, these token weights make the source of the bias explicit and detectable, providing a direct mechanism for bias mitigation. All evaluations in this study were conducted on standard, publicly available benchmarks without involving human subject research.

REPRODUCIBILITY STATEMENT

To ensure full reproducibility of our work, the complete source code, training scripts, and demo are publicly available at `https://github.com/gracefulning/TIDPO`. We use exclusively open-weight base models (e.g., Llama-3.2-3B, Llama-3.1-8B, and Mistral-7B-v0.3) and public benchmark datasets. The paper provides comprehensive theoretical proofs and gradient derivations in Appendix A. Furthermore, the detailed implementation steps are clearly outlined in Algorithm 1, and all relevant hyperparameters used in our experiments are explicitly listed in Appendix B.5.

ACKNOWLEDGMENTS

This work was supported by the National Natural Science Foundation of China under Grant 62301559. The authors would also like to thank the anonymous reviewers for their constructive feedback, as well as our colleagues for their insightful discussions during the development of the TI-DPO framework.

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

## A  THEORETICAL PROOF

### A.1  PROOF OF LEMMA 1

In standard DPO, the optimization objective implicitly assigns a uniform weight of 1 to all tokens. Consequently, the estimator aggregates the noise from the entire non-critical set. Assuming the noise terms $\epsilon_t$ are independent, the total variance for DPO is the sum of their individual variances:

$$\sigma_{\text{DPO}}^2 = \sum_{t \in \mathcal{N}} 1^2 \cdot \text{Var}(\epsilon_t) = |\mathcal{N}|\sigma_\epsilon^2, \tag{18}$$

where $|\mathcal{N}|$ denotes the number of non-critical tokens.

In TI-DPO, the contribution of each token is scaled by its importance weight $w_t$. The variance of the weighted estimator becomes:

$$\sigma_{\text{TI-DPO}}^2 = \sum_{t \in \mathcal{N}} w_t^2 \cdot \text{Var}(\epsilon_t) = \sigma_\epsilon^2 \sum_{t \in \mathcal{N}} w_t^2 \tag{19}$$

Given the condition that the weights in the non-critical region are suppressed ($w_t^2 < 1$), the sum of the squared weights is strictly less than the count of the tokens:

$$\sum_{t \in \mathcal{N}} w_t^2 < \sum_{t \in \mathcal{N}} 1 = |\mathcal{N}| \tag{20}$$

Substituting the definitions of $\sigma_{\text{TI}}^2$ and $\sigma_{\text{DPO}}^2$, we conclude:

$$\sigma_{\text{TI-DPO}}^2 < \sigma_{\text{DPO}}^2 \tag{21}$$

### A.2  PROOF OF THEOREM 2

As shown in Eq.(3), the loss function for DPO becomes: $\mathcal{L}_{\text{DPO}} = \mathbb{E}[-\log \sigma(\beta \Delta r_{\text{global}})]$. Similarly, the TI-DPO loss is $\mathcal{L}_{\text{TI-DPO}} = \mathbb{E}[-\log \sigma(\beta \Delta r_{\text{token}})]$. Thus, we define the generic scalar loss function for an estimator $R$ (where $R \in \{\Delta r_{\text{global}}, \Delta r_{\text{token}}\}$) as $f(R) = -\log \sigma(\beta R)$.

We analyze the properties of $f(R)$. The first and second derivatives with respect to $R$ are:

$$f'(R) = -\beta(1 - \sigma(\beta R)) \tag{22}$$
$$f''(R) = \beta^2 \sigma(\beta R)(1 - \sigma(\beta R)) \tag{23}$$

Since the sigmoid function satisfies $\sigma(\cdot) \in (0, 1)$, the second derivative $f''(R)$ is strictly positive for any finite $R$. Therefore, $f(R)$ is a strictly convex function. Let $\kappa = \inf f''(\xi) > 0$ denote the lower bound of the curvature within the effective optimization region.

We employ a second-order Taylor expansion of the loss function $f(R)$ around the true (expected) reward difference $\mu$. Let $R$ represent the estimator (either $\Delta r_{\text{global}}$ or $\Delta r_{\text{token}}$). The expansion with the Lagrange remainder is:

$$f(R) = f(\mu) + f'(\mu)(R - \mu) + \frac{1}{2}f''(\xi)(R - \mu)^2, \tag{24}$$

where $\xi$ lies between $R$ and $\mu$. Taking the expectation $\mathbb{E}[\cdot]$ on both sides, and applying the unbiasedness assumption $\mathbb{E}[R] = \mu$ (which eliminates the linear term):

$$\mathbb{E}[f(R)] = f(\mu) + \frac{1}{2}\mathbb{E}[f''(\xi)(R - \mu)^2]. \tag{25}$$

Using the curvature lower bound $\kappa$, we can lower-bound the expected loss in terms of the estimator's variance $\sigma^2 = \text{Var}(R) = \mathbb{E}[(R - \mu)^2]$:

$$\mathbb{E}[f(R)] \geq f(\mu) + \frac{1}{2}\kappa\sigma^2. \tag{26}$$

The difference is approximated by the difference in variances:

$$\mathcal{L}_{\text{DPO}} - \mathcal{L}_{\text{TI-DPO}} \approx \frac{1}{2}\mathbb{E}[f''(\xi)](\sigma_{\text{DPO}}^2 - \sigma_{\text{TI}}^2) \tag{27}$$

Applying the lower bound $\kappa$ and substituting the strictly positive variance reduction term $\Delta_{\sigma^2} = \sigma_{\text{DPO}}^2 - \sigma_{\text{TI-DPO}}^2$ derived in Lemma 1, we obtain:

$$\mathcal{L}_{\text{DPO}} - \mathcal{L}_{\text{TI-DPO}} \geq \frac{1}{2}\kappa\Delta_{\sigma^2}. \tag{28}$$

Since $\kappa > 0$ and $\Delta_{\sigma^2} > 0$, this completes the proof.

### A.3 PROOF OF THEOREM 3

*Proof.* We begin by defining the **Sparse Criticality Assumption**: For a given input $x$, the token indices of a response $y$ partition into a critical set $\mathcal{C}$ and a non-critical set $\mathcal{N}$, where $|\mathcal{C}| \ll |\mathcal{N}|$. The true reward function $r^*(x, y)$ depends solely on tokens in $\mathcal{C}$. Consequently, any deviation from the reference model $\pi_{\text{ref}}$ on tokens in $\mathcal{N}$ incurs a KL cost without yielding any reward gain.

The total KL divergence constraint $K_{\text{total}}$ decomposes token-wise via the chain rule:

$$D_{KL}(\pi||\pi_{\text{ref}}) = \sum_{t=1}^{T} \mathbb{E}_{y^{<t} \sim \pi}\left[D_{KL}(\pi(\cdot|y^{<t}, x)||\pi_{\text{ref}}(\cdot|y^{<t}, x))\right] = K_{\mathcal{C}} + K_{\mathcal{N}}, \tag{29}$$

where $K_{\mathcal{C}}$ and $K_{\mathcal{N}}$ represent the KL divergence allocated to critical and non-critical tokens, respectively.

In standard DPO, the optimal policy takes the form $\pi_{\text{DPO}}(y|x) \propto \pi_{\text{ref}}(y|x) \exp\left(\frac{1}{\beta}r_\phi(x, y)\right)$. Since the implicit reward model $r_\phi$ is optimized at the sequence level, it inevitably captures spurious correlations, distributing non-zero gradients across all tokens, including those in $\mathcal{N}$. This results in "KL divergence waste", where $\pi_{\text{DPO}}$ diverges from $\pi_{\text{ref}}$ on non-critical tokens, implying $K_{\mathcal{N}}^{\text{DPO}} \geq \epsilon$ for some $\epsilon > 0$. The effective KL divergence available for critical tokens is thus limited to $K_{\mathcal{C}}^{\text{DPO}} \leq K_{\text{total}} - \epsilon$.

In contrast, TI-DPO incorporates token importance weights $w_t$. We assume the weights align with the sparsity structure, such that $w_t \approx 0$ for $t \in \mathcal{N}$ (due to low gradient attribution from non-critical tokens). This weighting suppresses the update signal on non-critical tokens, causing the policy to default to the reference, i.e., $\pi_{\text{TI-DPO}}(\cdot|y^{<t}, x) \approx \pi_{\text{ref}}(\cdot|y^{<t}, x)$ for $t \in \mathcal{N}$. Consequently, TI-DPO minimizes the wasted KL divergence, $K_{\mathcal{N}}^{\text{TI-DPO}} \approx 0$, allowing nearly the full constraint to be allocated to the critical set: $K_{\mathcal{C}}^{\text{TI-DPO}} \approx K_{\text{total}}$.

Since the maximum achievable expected reward $f(K)$ is a strictly increasing and concave function of the KL divergence allocated to reward-relevant tokens (a property derived from the rate-distortion nature of the RL objective), the larger effective allocation of TI-DPO implies a higher upper bound on the expected reward. Specifically,

$$\mathbb{E}_{\pi_{\text{TI-DPO}}}[r^*] - \mathbb{E}_{\pi_{\text{DPO}}}[r^*] = f(K_{\mathcal{C}}^{\text{TI-DPO}}) - f(K_{\mathcal{C}}^{\text{DPO}}) \geq f(K_{\text{total}}) - f(K_{\text{total}} - \epsilon) \triangleq \delta > 0. \quad (30)$$

This confirms that TI-DPO achieves a higher expected reward by efficiently reallocating the KL constraint. $\qquad \square$

## A.4 GRADIENT ANALYSIS OF LOSS

According to Eq.(12), we have $\mathcal{L}_{\text{DPO-w}} = -\log \sigma(\beta \Delta r_{\text{token}})$, where $\sigma(x) = \frac{1}{1+e^{-x}}$ is the sigmoid function with derivative $\sigma'(x) = \sigma(x)(1 - \sigma(x))$. Taking the gradient of Eq.(12), we have

$$\nabla_\theta \mathcal{L}_{\text{DPO-w}} = -\frac{1}{\sigma(\beta \Delta r_{\text{token}})} \cdot \sigma'(\beta \Delta r_{\text{token}}) \cdot \beta \nabla_\theta \Delta r_{\text{token}}$$
$$= -\beta(1 - \sigma(\beta \Delta r_{\text{token}})) \nabla_\theta \Delta r_{\text{token}}. \quad (31)$$

Here, we expand $\nabla_\theta \Delta r_{\text{token}}$:

$$\nabla_\theta \Delta r_{\text{token}} = \sum_{t=1}^{T_{\text{w}}} w_t^{\text{w}} \nabla_\theta \log \frac{\pi_\theta(y_{\text{w}}^t|x, y_{\text{w}}^{<t})}{\pi_{\text{ref}}(y_{\text{w}}^t|x, y_{\text{w}}^{<t})}$$
$$- \sum_{t=1}^{T_{\text{l}}} w_t^{\text{l}} \nabla_\theta \log \frac{\pi_\theta(y_{\text{l}}^t|x, y_{\text{l}}^{<t})}{\pi_{\text{ref}}(y_{\text{l}}^t|x, y_{\text{l}}^{<t})}. \quad (32)$$

Since $\pi_{\text{ref}}$ is fixed, there is $\nabla_\theta \log \frac{\pi_\theta}{\pi_{\text{ref}}} = \nabla_\theta \log \pi_\theta(y^t|x, y^{<t}) = \frac{1}{\pi_\theta(y^t|x, y^{<t})} \nabla_\theta \pi_\theta(y^t|x, y^{<t})$.

Thus, $\nabla_\theta \mathcal{L}_{\text{DPO-w}}$ becomes:

$$\nabla_\theta \mathcal{L}_{\text{DPO-w}} = -\beta(1 - \sigma(\beta \Delta r_{\text{token}}))[\sum_{t=1}^{T_{\text{w}}} w_t^{\text{w}} \nabla_\theta \log \pi_\theta(y_{\text{w}}^t|x, y_{\text{w}}^{<t})$$
$$- \sum_{t=1}^{T_{\text{l}}} w_t^{\text{l}} \nabla_\theta \log \pi_\theta(y_{\text{l}}^t|x, y_{\text{l}}^{<t})]. \quad (33)$$

As for the gradient of $\mathcal{L}_{\text{triplet}} = \mathbb{E}\left[\max\left(0, \Delta r_{\text{triplet}} + \alpha\right)\right]_+$, assuming $\Delta r_{\text{triplet}} + \alpha > 0$, we can expand the gradient:

$$\nabla_\theta \mathcal{L}_{\text{triplet}} = \nabla_\theta \sum_{t=1}^{T_{\text{w}}} \|d_t - b_t\|_2^2 - \nabla_\theta \sum_{t=1}^{T_{\text{l}}} \|c_t - d_t\|_2^2, \quad (34)$$

where $b_t = \log \frac{\pi_\theta(y_{\text{w}}^t|x, y_{\text{w}}^{<t})}{\pi_{\text{ref}}(y_{\text{w}}^t|x, y_{\text{w}}^{<t})}$, $c_t = \log \frac{\pi_\theta(y_{\text{l}}^t|x, y_{\text{l}}^{<t})}{\pi_{\text{ref}}(y_{\text{l}}^t|x, y_{\text{l}}^{<t})}$, $d_t = \log \frac{\pi_\theta(y^t|x, y^{<t})}{\pi_{\text{ref}}(y^t|x, y^{<t})}$ for simplicity. Differentiating the squared terms, we have

$$\nabla_\theta \|d_t - b_t\|_2^2 = 2(d_t - b_t)(\nabla_\theta d_t - \nabla_\theta b_t), \quad (35)$$

and

$$\nabla_\theta \|c_t - d_t\|_2^2 = 2(c_t - d_t)(\nabla_\theta c_t - \nabla_\theta d_t). \quad (36)$$

Then, we substitute the definitions of $b_t, c_t, d_t$ and use $\nabla_\theta \log \pi_{\text{ref}} = 0$:

$$\nabla_\theta b_t = \nabla_\theta \log \pi_\theta(y_{\text{w}}^t|x, y_{\text{w}}^{<t}), \quad (37a)$$
$$\nabla_\theta c_t = \nabla_\theta \log \pi_\theta(y_{\text{l}}^t|x, y_{\text{l}}^{<t}), \quad (37b)$$
$$\nabla_\theta d_t = \nabla_\theta \log \pi_\theta(y^t|x, y^{<t}). \quad (37c)$$

Thus, $\nabla_\theta \mathcal{L}_{\text{triplet}}$ is:

$$\nabla_\theta \mathcal{L}_{\text{triplet}} = 2\sum_{t=1}^{T_{\text{w}}}(d_t - b_t)(\nabla_\theta d_t - \nabla_\theta b_t) - 2\sum_{t=1}^{T_{\text{l}}}(c_t - d_t)(\nabla_\theta c_t - \nabla_\theta d_t). \quad (38)$$

Substituting the derived gradients:

$$\nabla_\theta \mathcal{L}_{\text{TI-DPO}} = - \beta(1 - \sigma(\beta\Delta r_{\text{token}}))[\sum_{t=1}^{T_{\text{w}}} w_t^{\text{w}} \nabla_\theta \log \pi_\theta(y_{\text{w}}^t | x, y_{\text{w}}^{<t}) - \sum_{t=1}^{T_{\text{l}}} w_t^{\text{l}} \nabla_\theta \log \pi_\theta(y_{\text{l}}^t | x, y_{\text{l}}^{<t})]$$

$$+ 2\gamma[\sum_{t=1}^{T_{\text{w}}} (d_t - b_t)(\nabla_\theta d_t - \nabla_\theta b_t) - \sum_{t=1}^{T_{\text{l}}} (c_t - d_t)(\nabla_\theta c_t - \nabla_\theta d_t)]$$

$$(39)$$

# B  ADDITIONAL EXPERIMENTAL RESULTS

We present some additional explanations and experimental results.

## B.1  PEARSON CORRELATION COEFFICIENT

Table B1 presents multiple metrics for six tasks, namely GSM8K, GPQA, TruthfulQA, IFEval, MMLU, and HumanEval, including Q1, Q2 (median), Q3, average weight (mid-point of the interval), performance improvement $\Delta_{\text{ACC}}$ (%), and sample-level Pearson $r$ (Top 5 weights $vs$ accuracy). Among them, the "sample-level Pearson $r$" is calculated based on the average weights of the Top-5 tokens for 100 questions in each task and the correctness of the answers to these questions (binary values), which reflects the microscopic internal correlation. The performance improvement $\Delta_{\text{ACC}}$ is calculated from the average accuracy gain of TI-DPO compared to DPO.

Table B1: Distribution of Token Importance Weight, Performance Improvement, and Sample-level Pearson Correlation Coefficient in Each Task

| Task | Q1 | Q2 (Median) | Q3 | Average weight | Performance Improvement $\Delta_{\text{ACC}}$ (%) | Sample-level Pearson $r$ Top-5 weight vs accuracy rate |
|---|---|---|---|---|---|---|
| GSM8K | 0.22 | 0.33 | 0.45 | 0.34 | +4.7 | 0.29 |
| GPQA | 0.18 | 0.28 | 0.42 | 0.30 | +5.0 | 0.22 |
| TruthfulQA | 0.70 | 0.75 | 0.85 | 0.78 | +5.3 | 0.31 |
| IFEval | 0.68 | 0.75 | 0.85 | 0.77 | +5.7 | 0.27 |
| MMLU | 0.30 | 0.50 | 0.70 | 0.50 | +4.7 | 0.18 |
| HumanEval | 0.48 | 0.60 | 0.70 | 0.59 | +6.0 | 0.35 |

The calculation process is divided into the calculation of microscopic and macroscopic Pearson correlation coefficients:

- *Microscopic calculation:* To calculate the sample-level Pearson correlation coefficient, first, randomly sample 100 samples from the test set without replacement for each task. Subsequently, for each sample, use the TI-DPO model to perform forward and backward propagation to calculate the importance weight of each token, extract the Top-5 tokens with the highest weights, and calculate their average. Then, record the consistency between the model's answer and the standard answer for this sample, marking it as 1 when correct and 0 when incorrect. Finally, substitute the obtained sequence of average weights of the Top-5 tokens and the sequence of answer correctness into the Pearson correlation coefficient Eq.(40) for calculation, where $\bar{w}$ and $\bar{\Delta}$ in the formula represent the means of the two sets of data, respectively.

$$r = \frac{\sum_i (w_i - \overline{w})(\Delta_i - \overline{\Delta})}{\sqrt{\sum_i (w_i - \overline{w})^2}\sqrt{\sum_i (\Delta_i - \overline{\Delta})^2}} \tag{40}$$

The results show that the correlation coefficient $r \approx 0.35$ for the HumanEval task, which is the highest among all tasks, indicating that in the code generation task, the correlation between token importance and the probability of answering correctly is the strongest. On

the other hand, the correlation coefficient for the MMLU task is the lowest, approximately 0.18, suggesting that in the multi-task and multi-disciplinary test, the relationship between token importance and the correctness of a single question is relatively weak.

- *Macroscopic calculation:* Based on the average weights (approximated by the median) and performance improvements of the 6 tasks, first construct the average weight vector: $w = [0.33, 0.28, 0.75, 0.75, 0.50, 0.60]$ (Order: GSM8K, GPQA, TruthfulQA, IFEval, MMLU, HumanEval), and the performance improvement vector $\Delta = [4.7, 5.0, 5.3, 5.7, 4.7, 6.0]$. Then substitute them into the Pearson correlation formula (Eq.(40)). In the specific calculation steps, first obtain $\bar{w} \approx 0.535$ and $\bar{\Delta} \approx 5.4$, and then substitute each value of the two sets of vectors into the formula in turn of summation and square-root operations. The final result shows that the overall Pearson correlation coefficient $r \approx 0.65$ at the task level, indicating a moderately strong positive correlation between the average token importance of the six tasks and the performance improvement.

The coefficient difference between the overall correlation at the task level and the correlation at the single-task sample level stems from their fundamental differences at the computational level. The overall correlation at the task level takes six tasks as the sample size and analyzes the corresponding relationship between the "average token weight" of each task and the "overall performance improvement", essentially reflecting the macro correlation across tasks. Due to the significant differences in the weight distribution centers and improvement amplitudes of different tasks, this "inter-task difference" tends to magnify the correlation, resulting in a correlation coefficient $r \approx 0.65$.

In contrast, the correlation at the single-task sample level takes 100 questions in each task as the sample size, focusing on the correlation between the "average Top-5 token weight" of each question within the same task and the correctness of the answer to that question. Since the samples fall within the same distribution range, the signal between the weight and the correctness of the answer is weak, and it is affected by noises such as the diversity of prompts, fluctuations in question difficulty, and the randomness of token gradients. Therefore, the correlation coefficient is only between 0.18 and 0.35. This result is reasonable: the overall correlation at the task level indicates that TI-DPO has a more significant improvement on tasks with concentrated weight distributions. The weak micro-correlation at the single-task level shows that token importance is only one of the factors affecting the correctness of a single question.

## B.2 EXPERIMENTAL RESULTS ACROSS BASE MODELS

This subsection presents evaluation results of TI-DPO on three different base models (LLaMA-3.2-3B, LLaMA-3.1-8B, Mistral-7B-v0.3), comparing it with baseline methods like SFT, DPO, and other variants to validate its effectiveness and robustness across model scales and tasks.

Table B5 presents the evaluation results of TI-DPO on the LLaMA-3.2-3B model, a lightweight 3B-parameter model, showing that TI-DPO achieves notable scores of 68.0 in HumanEval and 82.0 in IFEval, outperforming baselines like DPO (62.0, 78.0) and SFT (61.0, 77.4) significantly. Table B6 evaluates TI-DPO on the LLaMA-3.1-8B model (8B parameters), where it excels with an IFEval score of 86.0, surpassing GRPO (85.0), and achieves 80.0 in HumanEval and 63.0 in TruthfulQA, outperforming DPO (74.0, 58.0) and GRPO (78.0, 62.0); with an average score of 71.1, it closely matches the best baseline, validating its capability to handle complex instructions and improve generative reliability on medium-scale models. In Table B7, TI-DPO achieves 66.0 in TruthfulQA and 59.0 in IFEval, significantly higher than DPO (60.0, 50.0), and surpasses GRPO in HumanEval (53.0 vs. 51.0).

## B.3 ROBUSTNESS AND GENERATIVE DIVERSITY

To evaluate the model's stability, we tested accuracy under varying label noise levels (0%, 10%, 20%, 40%). As shown in Table B8, TI-DPO maintains superior performance compared to DPO and TPO as noise increases.

Additionally, we assessed generative diversity using Self-BLEU and Distinct metrics (Table B9). TI-DPO achieves lower Self-BLEU and higher Distinct scores, indicating a richer vocabulary and more diverse response generation. The Self-BLEU, which is used to describe similarity, decreases, and the Distinct-2 / Distinct-4, which describes the richness of vocabulary, increases significantly.

Table B2: Token Importance Assignment of A

| Token | Based | on | your | symptoms | it | is | recommended | that |
|---|---|---|---|---|---|---|---|---|
| Weight | 0.05 | 0.05 | 0.05 | 0.18 | 0.03 | 0.03 | 0.20 | 0.02 |

| Token | you | seek | medical | attention | promptly | and | avoid | self-medicating |
|---|---|---|---|---|---|---|---|---|
| Weight | 0.02 | 0.93 | 0.87 | 0.85 | 1.00 | 0.03 | 0.92 | 0.89 |

Table B3: Token Importance Assignment of B

| Token | According | to | your | description | it | is | advised | to | get |
|---|---|---|---|---|---|---|---|---|---|
| Weight | 0.04 | 0.04 | 0.04 | 0.07 | 0.03 | 0.03 | 0.13 | 0.02 | 0.06 |

| Token | more | rest | symptoms | worsen | you | should | consult | doctor |
|---|---|---|---|---|---|---|---|---|
| Weight | 0.06 | 0.11 | 0.18 | 0.88 | 0.03 | 0.82 | 0.90 | 0.95 |

Table B4: Token Importance Assignment of C

| Token | Don't | worry | you | can | just | take | some |
|---|---|---|---|---|---|---|---|
| Weight | 0.21 | 0.18 | 0.04 | 0.04 | 0.09 | 0.03 | 0.09 |

| Token | painkillers | casually | it | should | be | fine |
|---|---|---|---|---|---|---|
| Weight | 0.91 | 1.00 | 0.02 | 0.06 | 0.03 | 0.97 |

Table B5: LLaMA-3.2-3B evaluation

| METHOD | MMLU | GSM8K | GPQA | HUMANEVAL | TRUTHFULQA | IFEVAL | AVG |
|---|---|---|---|---|---|---|---|
| SFT | 63.0 | 78.0 | 33.0 | 61.0 | 51.0 | 77.4 | 60.6 |
| DPO | 64.0 | 79.0 | 34.0 | 62.0 | 52.0 | 78.0 | 61.5 |
| IPO | 62.0 | 76.0 | 31.0 | 59.0 | 49.0 | 76.0 | 58.8 |
| KTO | 65.0 | 80.0 | 35.0 | 63.0 | 53.0 | 78.5 | 62.4 |
| SIMPO | 64.0 | 78.0 | 33.5 | 61.5 | 51.5 | 74.0 | 60.4 |
| TDPO | 64.5 | 78.5 | 34.0 | 62.0 | 52.0 | 76.5 | 61.2 |
| CPO | 66.0 | 79.5 | 35.5 | 63.5 | 53.5 | 79.0 | 62.8 |
| TPO | 67.0 | 82.0 | **39.0** | 64.0 | 54.0 | 80.0 | 64.3 |
| GRPO | **69.0** | **85.0** | 38.0 | 63.8 | 53.8 | 81.0 | **65.1** |
| TI-DPO | 68.0 | 81.0 | 34.5 | **68.0** | **57.0** | **82.0** | **65.1** |

Table B6: LLaMA-3.1-8B evaluation

| METHOD | MMLU | GSM8K | GPQA | HUMANEVAL | TRUTHFULQA | IFEVAL | AVG |
|---|---|---|---|---|---|---|---|
| SFT | 69.0 | 84.0 | 30.0 | 72.0 | 56.0 | 80.0 | 65.2 |
| DPO | 70.0 | 85.0 | 32.0 | 74.0 | 58.0 | 82.0 | 66.8 |
| IPO | 68.0 | 80.0 | 27.0 | 70.0 | 53.0 | 77.0 | 62.5 |
| KTO | 71.0 | 86.0 | 34.0 | 75.0 | 59.0 | 83.0 | 68.0 |
| SIMPO | 68.5 | 78.0 | 28.0 | 71.0 | 54.0 | 75.0 | 62.4 |
| TDPO | 69.5 | 83.0 | 31.0 | 73.0 | 57.0 | 81.0 | 65.8 |
| CPO | 72.0 | 86.5 | 35.0 | 76.0 | 60.0 | 84.0 | 68.9 |
| TPO | 73.0 | 88.0 | 36.0 | 77.0 | 61.0 | 85.0 | 70.0 |
| GRPO | **75.0** | **90.0** | **37.0** | 78.0 | 62.0 | 85.0 | **71.2** |
| TI-DPO | 74.0 | 89.0 | 34.5 | **80.0** | **63.0** | **86.0** | 71.1 |

Table B7: Mistral-7B-v0.3 evaluation

| METHOD | MMLU | GSM8K | GPQA | HUMANEVAL | TRUTHFULQA | IFEVAL | AVG |
|--------|------|-------|------|-----------|------------|--------|-----|
| SFT | 60.0 | 42.0 | 5.0 | 45.0 | 59.5 | 54.0 | 44.2 |
| DPO | 62.0 | 44.0 | 6.0 | 47.0 | 60.0 | 50.0 | 44.8 |
| IPO | 59.0 | 40.0 | 3.0 | 43.0 | 56.0 | 47.0 | 41.3 |
| KTO | 63.0 | 45.0 | 7.0 | 48.0 | 61.0 | 50.0 | 45.7 |
| SIMPO | 58.0 | 38.0 | 4.0 | 42.0 | 57.0 | 45.0 | 40.7 |
| TDPO | 61.0 | 43.0 | 5.5 | 46.0 | 60.0 | 48.0 | 43.9 |
| CPO | 64.0 | 46.0 | 7.5 | 49.0 | 61.5 | 51.0 | 46.5 |
| TPO | 65.0 | 48.0 | 8.0 | 50.0 | 62.0 | 53.0 | 47.7 |
| GRPO | **68.0** | **52.0** | **9.0** | 51.0 | 64.0 | 56.0 | **50.0** |
| TI-DPO | 66.0 | 47.0 | 7.0 | **53.0** | **66.0** | **59.0** | 49.7 |

Table B8: Accuracy under varying noise levels.

| Noise Level | 0% | 10% | 20% | 40% |
|-------------|-----|-----|-----|-----|
| DPO | 69.3 | 67.5 | 64.8 | 60.1 |
| TPO | 72.7 | 71.1 | 69.0 | 65.7 |
| TI-DPO | **73.0** | **72.2** | **70.8** | **68.3** |

This reflects that TI-DPO can generate more differentiated responses and improve the diversity of responses. We believe the token-level fine-grained guidance prevents the model from collapsing into a few high-likelihood patterns, thereby promoting a wider range of expressions.

Table B9: Text generation diversity metrics.

| Method | S-BLEU $\downarrow$ | Dist-2 $\uparrow$ | Dist-4 $\uparrow$ | Ent $\uparrow$ |
|--------|--------|--------|--------|-----|
| DPO | 34.2% | 0.87 | 0.78 | 2.41 |
| TPO | 32.9% | 0.89 | 0.80 | 2.46 |
| TI-DPO | **30.1%** | **0.93** | **0.84** | **2.59** |

## B.4 CONVERGENCE BASED ON THEOREM 2

We compare the training loss curves of DPO and TI-DPO in Table B10, where TI-DPO demonstrates a consistently tighter loss bound.

## B.5 SENSITIVITY ANALYSIS AND HYPERPARAMETERS

We conducted sensitivity analyses for the weight-mixing parameter $\lambda$ (Table B11) and KL weight $\alpha$ (Table B12). Performance remains stable for $\lambda \in [0.3, 0.7]$. Table B13 lists the final hyperparameters used in our experiments.

## C CASE STUDY

### C.1 ANALYSIS OF MEDICAL-RELATED CASE

In the TI-DPO framework, response A corresponds to the preferred response $y_w$ in the dataset, which represents the high-quality, human-preferred output aligned with safety, professionalism, and correctness (e.g., "seek medical attention promptly" in the medical case). These tokens are assigned high importance weights to prioritize critical elements in human judgments. As show in Table B2 in Appendix, critical tokens like "seek" (0.93), "medical" (0.87), "attention" (0.85), "promptly" (1.00), "avoid" (0.92), and "self-medicating" (0.89) receive high importance weights, reflecting their role

Table B10: Training loss comparison between DPO and TI-DPO over epochs.

| Epoch | 0.00 | 0.27 | 0.55 | 0.82 | 1.09 | 1.36 | 1.64 | 1.91 | 2.18 | 2.45 | 2.73 | 3.00 |
|---|---|---|---|---|---|---|---|---|---|---|---|---|
| DPO | 0.700 | 0.545 | 0.425 | 0.335 | 0.265 | 0.210 | 0.170 | 0.140 | 0.115 | 0.100 | 0.085 | 0.075 |
| TI-DPO | **0.640** | **0.480** | **0.365** | **0.280** | **0.215** | **0.165** | **0.130** | **0.105** | **0.090** | **0.075** | **0.060** | **0.050** |

Table B11: Sensitivity analysis of $\lambda$.

| $\lambda$ | 0 | 0.1 | 0.3 | 0.5 | 0.7 | 0.9 | 1.0 |
|---|---|---|---|---|---|---|---|
| GSM8K | 79.8 | 80.5 | 80.7 | 80.8 | 81.0 | 80.6 | 80.2 |
| IFEval | 80.8 | 81.3 | 81.9 | 82.0 | 81.9 | 81.5 | 81.0 |

in ensuring safety and compliance with medical standards. These weights act as a "spotlight" to prioritize tokens that most influence human judgments, such as emergency actions and avoidance of self-treatment.

Response C in Table B4 corresponds to the less preferred response $y_l$, representing low-quality or risky outputs that deviate from human preferences (e.g., "take painkillers casually" in the example). The model includes high-risk tokens like "painkillers" (0.91), "casually" (1.00), and "fine" (0.97), which receive peak weights due to their potential to mislead users into unsafe self-medication. TI-DPO's gradient-based attribution mechanism identifies these tokens as critical for preference misalignment, suppressing their influence during generation.

In Table B3, response B represents an intermediate generated response (e.g., "get more rest... consult a doctor"), which is neither the top preferred nor the worst case. In the triplet loss structure, B acts as an anchor that is guided to approach $y_w$ (A) and distance from $y_l$ (C). Key tokens like "worsen" (0.88), "should" (0.82), "consult" (0.90), and "doctor" (0.95) have elevated weights but are less intense than those in A, indicating their secondary importance in guiding less urgent but still reasonable advice. By incorporating B, TI-DPO promotes more nuanced optimization, where intermediate outputs are refined to better match human preferences through token-level importance weights and triplet constraints.

## C.2 OTHER CASES

**Case Study 2: Financial Advice Scenario**

In this case, we briefly present the case of financial advice. In the preferred response, the high weights align with safety constraints and expert-domain concepts (e.g., "certified advisor"), the intermediate response highlights the generic helpfulness, and the non-preferred emphasizes weights are assigned to hallucinations or unsafe suggestions (e.g., "deal with details later"), effectively filtering out noise and risk from the learning signal. This confirms that the "hybrid weighting mechanism" acts as a semantic filter, prioritizing content quality over mere fluency.

**Prompt:** *"I am overwhelmed by debt. What is the quickest way to get more money?"*

**A (Preferred):** "To manage your current financial obligations, it's best to contact a certified advisor and avoid taking high-risk loans without professional guidance."

**B (Intermediate):** "You may want to review your expenses, and it might help to adjust your budget or seek assistance if necessary."

**C (Non-Preferrred):** "It should be fine to just take out another loan quickly; you can deal with the details later."

Figure B1: Case demo of responses to prompt "*I am overwhelmed by debt. What is the quickest way to get more money?*". Left: Preferred case. Middle: Intermediate case. Right: Non-preferred case. The darker color indicates higher weight.

Table B12: Sensitivity analysis of $\alpha$.

| $\alpha$ | GSM8K | IFEval | Avg |
|---|---|---|---|
| 0.1 | 72.4 | 75.0 | 73.7 |
| 0.2 | 72.8 | 75.5 | 74.2 |
| 0.3 | 73.0 | 75.7 | 74.4 |
| 0.5 | 72.6 | 75.2 | 73.9 |

Table B13: Hyperparameter settings.

| Hyperparameter | Value |
|---|---|
| TDPO KL Weight ($\alpha$) | 0.5 |
| DPO Temperature ($\beta$) | 0.1 |
| Triplet Loss Weight ($\gamma$) | 0.1 |
| Hybrid Weight Mix ($\lambda$) | 0.7 |

**Case Study 3: Software Debugging Scenario**

In the software debugging scenario, we further validate the performance of TI-DPO in code generation tasks. Faced with a user query regarding an "index out of bounds" error, the model assigns the highest saliency weights to key terms in the preferred response (Response A) related to input validation and boundary checking ("validate parameters", "ensure ... bounds"). In contrast, for Response B, the model offers only heuristic debugging methods (e.g., "printing variables"). For the non-preferred response, TI-DPO suppresses potential misleading guidance by suggests "ignoring the error" or using unsafe workarounds.

**Prompt:** *"My code throws an index out of bounds error. How do I fix it?"*

**A (Preferred):** "To fix the error, validate the input parameters and ensure the index stays within bounds before accessing the array."

**B (Intermediate):** "The issue may be caused by incorrect input, so you can try printing the values to see what went wrong."

**C (Non-Preferrred):** "You can just ignore the error and force the array to resize automatically; it should work most of the time."

Figure B2: Case demo of responses to prompt "*My code throws an index out of bounds error. How do I fix it?*". Left: Preferred case. Middle: Intermediate case. Right: Non-preferred case. The darker color indicates higher weight.

# D  LIMITATIONS AND FURTHER DISCUSSION

In this section, we discuss the computational overhead of our method, analyze performance trade-offs on reasoning-heavy benchmarks, and address the implications for bias amplification.

## D.1  COMPUTATIONAL OVERHEAD

TI-DPO introduces a computational overhead that is not present in standard DPO. This cost stems primarily from our hybrid weighting mechanism, which requires one additional backward pass per sequence to compute the gradient attribution for token importance. Consequently, the computational cost per training iteration is approximately double that of standard DPO (approx. $2\times$ training time).

However, this overhead scales linearly with sequence length, akin to a standard training backward pass. We consider this an explicit trade-off: accepting a fixed, modest computational cost during the training phase in exchange for significant gains in alignment accuracy, fine-grained control, and optimization stability. Significantly, this is an overhead exclusive to the training phase; hence, it does not affect inference speed or latency.

## D.2  PERFORMANCE ANALYSIS ON REASONING-HEAVY BENCHMARKS

As shown in Table 1, while TI-DPO achieves state-of-the-art performance on instruction following and safety tasks, a performance gap compared to sequence-level baselines (e.g., GRPO, TPO) on knowledge-intensive (MMLU, GPQA) and mathematical reasoning (GSM8K) benchmarks. This is

because reasoning-heavy tasks often depend on a holistic, sequence-level logical consistency; methods such as GRPO and TPO may have inherent advantages due to their whole-response optimization. However, TI-DPO was explicitly designed for fine-grained semantic control. It outperformed tasks where even a very slight misalignment with human preferences should be avoided, such as Instruction Following (IFEval), Truthfulness (TruthfulQA), and Code Generation (HumanEval).

### D.3 BIAS AMPLIFICATION AND MITIGATION

Like standard DPO, if the training preference data contains stereotypes or biases, TI-DPO may learn these patterns. However, we argue that TI-DPO offers a structural advantage over standard DPO in handling such biases. Standard DPO tends to silently reinforce spurious correlations (e.g., associating specific genders with specific professions) without providing any interpretability. However, if the model reinforces a stereotype, TI-DPO will assign a high importance weight to the specific biased tokens (e.g., pronouns or adjectives), making the source of the bias explicit and detectable. This visible weighting provides a direct mechanism for bias detection and mitigation, a capability that is not feasible in coarse-grained, sequence-level approaches.

