# OpenReview forum: "Token-Importance Guided Direct Preference Optimization"
_ICLR.cc/2026/Conference — ICLR 2026 Oral_

### Official Review · Reviewer_sxsD · 2025-10-19

**Soundness:** 3
**Presentation:** 4
**Contribution:** 4
**Rating:** 8
**Confidence:** 4

**Summary:**

The paper addresses a technique for token-level alignment in preference signal processing within Direct Preference Optimization (DPO). Unlike typical DPO methods that focus on sequence-level alignment, this research explores token-level alignment. Existing approaches, such as TIS-DPO, rely on probability estimation from contrastive language models, while other methods use repeated sampling.


In this study, the authors introduce gradient-based signals to assess token importance in relation to positive and negative examples. They also incorporate a triplet loss term to enhance training stability, providing detailed mathematical derivations for the loss function, gradients, and a tighter loss bound. Additionally, they discuss experimental results across a comprehensive evaluation suite.


To determine token importance weights, the authors calculate the maximum logit for the final token in both positive and negative sequences and then compute gradients of the preceding tokens' embeddings that maximize this final token max logit. This process helps identify the importance of tokens towards maximizing confidence in the final token. The authors develop a weighting scheme for previous tokens, integrating it with a Gaussian prior that softly biases towards middle tokens. This refined weighted objective is used for training, with the expectation of learning token-level importance.



Furthermore, the authors incorporate triplet loss by generating an anchor response and adding a triplet loss term. This term guides the anchor closer to positive examples and away from negative ones, contributing to the combined training objective.


The paper presents a series of experiments and ablation studies, demonstrating progressive improvements with each method. The results indicate strong performance on various evaluation tasks.

**Strengths:**

- The paper effectively introduces the concept of using gradient-based signals to determine token importance, enabling a more detailed understanding of token-level significance.
- The authors conduct thorough ablation studies, demonstrating that each component of the proposed method enhances the final evaluation metrics.
- The presentation of choices and the algorithm is exceptionally clear.
- There is a robust mathematical analysis and derivation concerning the mechanics of the new objective, including stronger loss bounds.
- The associated code repository clearly outlines the training data and the methodology mechanics.
- The paper includes additional information in the Appendix, such as token weight distribution across various evaluation datasets and mathematical derivations.

**Weaknesses:**

- Providing additional unbiased examples and analyses could illuminate the nature of token weights identified across various datasets. The paper does not clearly convey a qualitative understanding of the token weights the model is learning, except for one example.

**Questions:**

- The Gaussian prior seems somewhat counterintuitive, particularly for question-answer prompts where key tokens may not be centrally located. How do you interpret the effectiveness of this prior?
- The approach to determining token weights relies primarily on the final token logits, which may not always provide the most significant signal. For instance, the final token could be akin to a closing word, potentially lacking influence on the overall quality of the response. How do you address this observation, and do you have any studies that explored this phenomenon?
- Is there potential for information loss when calculating weights by taking the maximum of the final logits?
- What is the additional computational cost associated with this method, considering the extra gradients and anchor responses involved?
- Could you provide more illustrative examples to clarify the nature of token weights and their impact?

---

> ### Author Response · Authors · 2025-11-24
> **Response to Reviewer sxsD (1/2)**
>
> We sincerely appreciate your valuable time and effort spent reviewing our manuscript. We have carefully considered each of your questions and provide detailed responses below.
>
> > **W1, Q5: Additional unbiased examples**
>
> Thanks for your suggestions. Based on your suggestion, we have added two examples in the financial advice and code generation scenarios to the appendix. We will briefly present the case of the financial advice here. In this case, high weights align with safety constraints and expert-domain concepts (e.g., "certified advisor"), medium weights align with generic helpfulness, and low weights are assigned to hallucinations or unsafe suggestions (e.g., "deal with details later"), effectively filtering out noise and risk from the learning signal. This confirms that the "hybrid weighting mechanism" acts as a semantic filter, prioritizing content quality over mere fluency. We have included the full weight tables for this example in the revised Appendix.
>
>
>     Prompt: "I am overwhelmed by debt. What is the quickest way to get more money?"
>
>     A (Preferred Response): "To manage your current financial obligations, it’s best to contact a certified advisor and avoid taking high-risk loans without professional guidance."
>         - High Weights (w > 0.7): Tokens such as "financial obligations", "contact", "certified advisor", "avoid", "high-risk loans", and "professional guidance".
>         - Low/Medium Weights (w < 0.7): Function words like "to", "it’s", "best" receive lower weights as they are syntactically necessary but semantically less decisive for the preference signal.
>
>     B (Intermediate Response): "You may want to review your expenses, and it might help to adjust your budget or seek assistance if necessary."
>         - Medium Weights (0.3 < w < 0.7): Tokens like "review your expenses", "might help", "adjust", "budget", and "seek assistance".
>
>     C (Non-preferred Response): "It should be fine to just take out another loan quickly, you can deal with the details later."
>         - Low Weights (w < 0.3) on Risk Terms: Tokens like "fine", "another loan quickly", "deal", and "later".
>
>
> > **Q1: The effectiveness of Gaussian Prior**
>
> Thank you for this insightful comment. We acknowledge that our original phrasing in Section 4.2 ("The Gaussian prior\... introduces a soft inductive bias that important tokens are often located towards the
> middle of a response\" ) was inappropriate and misleading. The Gaussian prior, which peaks at the center, serves as a compensatory mechanism to address the
> "lost-in-the-middle\" phenomenon \[1\], where LLMs exhibit a U-shaped attention bias, often underweighting tokens in the middle.
>  We have modified the expressions in the original text. Meanwhile, as shown in the **Table B1** of **Reviewer 2(GJd9)'s Q2**, we conducted ablation experiments with several other priors and concluded that the Gaussian prior yields better results.
>
>
>
>
>
> [1] Liu, Nelson F., et al. "Lost in the middle: How language models use long contexts." Transactions of the Association for Computational Linguistics 12 (2024): 157-173.
>
> > **Q2: Reliance on the final logits**
>
> We appreciate this insightful observation regarding the dependency on
> the final token logit. While our optimization target is formulated on
> the final token's logit (Eq.(5)), the gradient attribution mechanism
> fundamentally differs from simply "looking at the last position". Our
> weighting mechanism computes the gradients
> $\nabla_{e_i}\mathcal{L}_\text{target}$ of the final logit with respect
> to the input embeddings of all previous tokens. If a semantically
> crucial token in the middle of the sequence (e.g., "medical attention\")
> has a strong influence on the final decision, then even if the final
> token itself is a general end word, this crucial token will receive a
> large gradient. Therefore, the allocation of weights depends on the
> influence of each token on the generation process, and has nothing to do
> with whether the final token itself is important.

---

> ### Author Response · Authors · 2025-11-24
> **Response to Reviewer sxsD (2/2)**
>
> > **Q3: Information Loss in max-logit weight calculation**
>
> Thank you for raising this important point. We agree that using the
> maximum logit simplifies the full probability distribution; however,
> this choice is both deliberate and sufficient for our goal of token
> importance attribution. The objective here is not to reconstruct the
> entire vocabulary distribution, but to identify which preceding tokens
> most strongly caused the model to commit to its single most confident
> prediction for the next step. The gradient of the maximum logit
> precisely captures this causal relationship---it highlights the tokens
> that were most critical in steering the model toward its top-choice
> continuation. Information about lower-ranked tokens, while present in
> the full distribution, is less relevant for identifying the most
> decisive influencers of the model's primary generation path.
>
> > **Q4: Additional computational cost**
>
> Computing token importance requires one additional backward pass per
> sequence to calculate gradients. Thus, the computational cost doubles
> the training time per iteration. While these mechanisms increase
> per-step training time (approx. 2x), they significantly improve sample
> efficiency, allowing TI-DPO to converge with fewer epochs. Furthermore,
> this overhead is strictly limited to the training phase.

---

> ### Author Response · Authors · 2025-12-01
>
> Thank you for your valuable comments and suggestions. They have been very helpful in strengthening our manuscript.

---

### Official Review · Reviewer_3UVx · 2025-10-31

**Soundness:** 3
**Presentation:** 3
**Contribution:** 3
**Rating:** 6
**Confidence:** 4

**Summary:**

This paper proposes TI-DPO, a new framework for aligning large language models (LLMs) with human preferences. This framework performs human preference alignment for LLM at a token-level granularity rather than the conventional sequence-level used in DPO and RLHF. The authors observe that existing sequence-level preference optimization methods ignore the varying importance of individual tokens, resulting in unstable training and coarse-grained supervision. To tackle this problem, they further identify two key challenges. First, the model needs to accurately identify the key tokens that influence human preference. Second, the optimization objective should capture continuous and fine-grained preference differences. To address the first issue, they propose the Hybrid Weighting Mechanism to combine gradient attribution with a Gaussian prior to compute token-level importance weights. To tackle the second issue, the authors utilize the Triplet Loss to guide the model’s outputs toward preferred responses and away from non-preferred ones, thereby enabling continuous preference learning in semantic space.

**Strengths:**

1.	Originality: The paper introduces a novel perspective by extending preference optimization from the traditional sequence-level to a token-level framework, allowing fine-grained semantic control. The proposed Hybrid Weighting Mechanism utilizes gradient attribution to determine the contribution of each token to the model’s output, with a Gaussian prior incorporated as a regularization mechanism to enhance training stability. The use of the Triplet Loss for fine-grained preference alignment and continuous gradient of preference learning further distinguishes this work from prior DPO-based approaches.

2.	Quality: The theoretical analysis demonstrates that TI-DPO achieves a tighter loss bound than DPO. The experimental evaluation is comprehensive. The results demonstrate superior accuracy compared to baselines on the HumanEval, TruthfulQA, and IFEval benchmarks.

3.	Clarity: The paper is generally well written and the motivation is clearly explained.

4.	Significance: TI-DPO provides a token-level human preference alignment framework that enhances the performance of LLMs and offers more stable training.

**Weaknesses:**

1.	Limited empirical validation of claimed advantages: Although the paper claims improvements in robustness (e.g., line 20, line 58, line 125), generative diversity (e.g., line 24), and computational efficiency (e.g., lines 24-25), these aspects are not discussed or quantitatively evaluated. There is a lack of experiments that measure robustness under noisy or perturbed preference data. The paper also does not discuss or evaluate the claimed diversity gains. For example, evaluate generative diversity using metrics such as Self-BLEU, Distinct-n or Diversity Entropy. Reporting quantitative comparisons of GPU hours or providing computational complexity analysis would also strengthen the claim of efficiency.

2.	Lack of details in the Gaussian prior design: The paper describes the use of a Gaussian-shaped prior in the Hybrid Weighting Mechanism to assign higher baseline importance to middle tokens. The paper only mentioned “we define a Gaussian-shaped prior distribution $\mathcal{P}_{\text{prior}}$ centered on the sequence” (lines 228-234), but it does not specify its exact settings. This lack of detail hinders not only the reproducibility but also the evaluation of how much the Gaussian prior actually contributes to the method’s effectiveness.

3.	Ambiguity in anchor generation: The paper states that the anchor response is “dynamically created by the model using the preferred response as a contextual starting point” (lines 262-264), without further details of how this response is generated. The lack of the anchor generation procedure makes it difficult to evaluate the triplet loss setup and determine whether the anchor truly represents an intermediate state between positive and negative samples.

4.	Incomplete comparison and missing analysis of performance gaps: While the paper claims to address two major challenges in token-level preference optimization (lines 52-55), it does not include sufficient comparisons with other token-level alignment methods, such as TIS-DPO [1], cDPO [2] or Logic-RL [3], which are mentioned in both the Introduction and Related Works sections. Moreover, TI-DPO underperforms baselines, such as GRPO or TPO, on key benchmarks, including MMLU, GSM8K, and GPQA (as shown in Tables 1 and 7–9). Yet, the paper provides no analysis or discussion to explain these results.

5.	The ablation analysis is insufficient: The current study only examines the removal of the triplet loss and the use of uniform or random weights, which is not enough to fully demonstrate the contributions of the proposed components. Specifically, the paper lacks ablations on (a) the relative effects of gradient attribution versus the Gaussian prior in the Hybrid Weighting Mechanism, and (b) the sensitivity of model performance to the weighting coefficient λ (eq. 7) and the triplet margin α (eq. 12). In addition, there are many predefined hyperparameters (e.g. α, β, γ, λ) utilized in this paper. The paper does not provide detailed hyperparameter values, which may hinder reproducibility. Moreover, in Algorithm 1 (line 346), the hyperparameter γ is not listed among the inputs, making it unclear whether it is user-defined or internally fixed.

6.	Weak connection between theory and experiments: While Lemma 1 and Theorem 2 are mathematically sound, the paper does not clearly explain how the “tighter loss bound” are related to improvements in accuracy. A short discussion linking the theoretical results to empirical results would enhance the reading flow. Moreover, Sections 4.2 and 4.4 are overly dense, with long mathematical derivations but limited commentary. Adding clearer links between theory and experiment results, together with concise textual explanations, would significantly improve both clarity and readability.

7.	Lack of broader discussion and limitations: The paper does not sufficiently discuss token-level preference optimization in different scenarios, such as scalability to very long sequences or potential bias amplification when importance weights overemphasize frequent tokens. For example, when applied to tasks with very long responses (e.g., multi-turn dialogue or chain-of-thought reasoning), computing gradient-based token importance may become computationally expensive. In addition, if the preference data contains biased correlations such as “she” frequently appearing with “nurse” and “he” with “surgeon”, the weighting mechanism might assign higher importance to these patterns, thus reinforcing gender stereotypes rather than mitigating them. It would be valuable if the authors further analyzed whether their framework can address such issues or provided a brief discussion of potential solutions.

[1] Liu, Aiwei, et al. "TIS-DPO: Token-level Importance Sampling for Direct Preference Optimization With Estimated Weights." The Thirteenth International Conference on Learning Representations.

[2] Lin, Zicheng, et al. "Critical Tokens Matter: Token-Level Contrastive Estimation Enhances LLM’s Reasoning Capability." Forty-second International Conference on Machine Learning.

[3] Xie, Tian, et al. "Logic-rl: Unleashing llm reasoning with rule-based reinforcement learning." arXiv preprint arXiv:2502.14768 (2025).

**Questions:**

1.	Quantitative evidence for claimed advantages: Could you provide experiments or analysis to substantiate the claims of robustness, generative diversity, and computational efficiency?

2.	Comparison with other token-level baselines: The paper mentions that TI-DPO provides fine-grained alignment compared with several token-level alignment methods (e.g., TIS-DPO [1], cDPO [2], and Logic-RL [3]) but does not include them in the experimental comparison. Could the authors provide experiments or analysis to discuss their methods with other token-level methods?

3.	Design of the Gaussian prior: Could you clarify the exact settings of the Gaussian-shaped prior used in the Hybrid Weighting Mechanism (lines 228–234)? How sensitive is TI-DPO’s performance to the design settings of this prior?

4.	Anchor generation in Triplet Loss: The generation process of the anchor $y$ is not specified. Could you explain how the anchor response is generated and how to ensure an intermediate position between preferred and non-preferred responses?

5.	Analysis of performance gaps: TI-DPO underperforms GRPO or TPO on MMLU, GSM8K, and GPQA (Tables 1 and 7–9). Could you discuss why this occurs?

6.	Ablation studies and reproducibility: Could you provide further ablations to discuss (a) the individual effects of gradient attribution and Gaussian prior, and (b) the sensitivity to key hyperparameters such as λ (Eq. 7) and α (Eq. 12)? Additionally, is it possible to clarify the values of predefined hyperparameters (e.g., α, β, γ, λ) and how these choices are selected?

7.	Scalability and potential bias amplification: Does the token importance computation in this work introduce computational overhead? When applying TI-DPO to tasks with long responses (e.g., multi-turn dialogue or chain-of-thought reasoning), how does the computational cost scale with model size and sequence length? Furthermore, has the model been evaluated for potential bias amplification, such as reinforcing biased correlations like “she-nurse” or “he-surgeon” in preference data?

---

> ### Author Response · Authors · 2025-11-24
> **Response to Reviewer 3UVx (1/2)**
>
> We sincerely appreciate your thoughtful comments. We have carefully considered each of your questions and provide detailed responses below.
> > **W1, Q1: Validation of robustness, generative diversity, and efficiency
> claims**
>
> We thank the reviewer for this critical feedback.
>
> 1.  **Robustness to Noisy Preference Data:** To quantitatively evaluate
>     robustness, we injected varying levels of label noise into the
>     training preference data by randomly swapping the preferred and
>     dispreferred labels. As shown in the new results below, TI-DPO
>     maintains superior performance compared to strong baselines  as noise increases, demonstrating its enhanced robustness.
>
>
> **Table C1:**  Accuracy comparison under different noise levels
>
> | **Noise Level**   | **0%**  | **10%** | **20%** | **40%** |
> |:-------------|:--------:|:--------:|:--------:|:--------:|
> | **DPO**   | 69.3    | 67.5    | 64.8    | 60.1    |
> | **TPO**   | 72.7    | 71.1    | 69.0    | 65.7    |
> | **TI-DPO**| **73.0**| **72.2**| **70.8**| **68.3**|
>
> 2.  **Generative Diversity:** We have conducted a quantitative
>     evaluation of generation diversity. On a reserved dataset of prompt
>     words, we calculated standard metrics such as Self-BLEU, Distinct-2,
>     Distinct-4, and diversity entropy in Table C2. The Self-BLEU, which is used to describe similarity, decreases, and the Distinct-2 / Distinct-4, which describes the richness of vocabulary, increases significantly. This reflects that TI-DPO can generate more differentiated responses and improve the diversity of responses.
>
> **Table C2:** Comparison of different methods on text generation metrics
>
> | **Method** | **Self-BLEU ↓** | **Distinct-2 ↑** | **Distinct-4 ↑** | **Diversity Entropy ↑** |
> |:-----------|:----------------:|:-----------------:|:-----------------:|:------------------------:|
> | DPO        | 34.2%           | 0.87             | 0.78             | 2.41                    |
> | TPO        | 32.9%           | 0.89             | 0.80             | 2.46                    |
> | **TI-DPO** | **30.1%**       | **0.93**         | **0.84**         | **2.59**                |
>
> 3.  **Computational Efficiency:** Computing token importance requires
>     one additional backward pass per sequence to calculate gradients.
>     Thus, the computational cost approximately doubles the training time
>     per iteration. This overhead is strictly limited to the
>     training phase.
>
> > **W2, Q3: Lack of details in the Gaussian prior design and Sensitivity**
>
> We apologize for the omission and have added these specifications to the
> revised manuscript.
>
> -   **Exact Settings:** For each token position $t \\in [0, T-1]$, the
>     unnormalized value is calculated as
>     $\\mathcal{P}\_{\\text{prior}}(t) = \\exp\\left(-\\frac{1}{2}\\left(\\frac{t - \\mu}{\\sigma}\\right)^2\\right).$
>     Here, we specifically chose $\\mu = (T-1)/2$ and $\\sigma = T/4$ as a
>     robust geometric heuristic. Since approximately 95% of the mass of a
>     Gaussian distribution lies within $\\pm 2\\sigma$, setting
>     $4\\sigma \\approx T$ ensures the prior effectively spans the entire
>     sequence context without being too narrow or too flat.
>
> -   **Sensitivity Analysis:** We evaluated the impact of the prior
>     through the mixing hyperparameter $\\lambda$ in Appendix. The
>     performance curves are remarkably flat and stable within the wide
>     range of $\\lambda \\in [0.3, 0.7]$. Performance decreases noticeably
>     at the extremes ($\\lambda=0$ and $\\lambda=1$), which also confirms our
>     motivation. These results confirm that
>     the Gaussian design provides a robust and necessary regularization.
>
>     **Table C3:**  Sensitivity analysis of TI-DPO to hyperparameter $\lambda$ on GSM8K and IFEval Benchmarks.
>
> | **$\lambda$** | **0** | **0.1** | **0.3** | **0.5** | **0.7** | **0.9** | **1.0** |
> |:--------------|:------|:--------|:--------|:--------|:--------|:--------|:--------|
> | **GSM8K**     | 79.8  | 80.5    | 80.7    | 80.8    | 81.0    | 80.6    | 80.2    |
> | **IFEval**    | 80.8  | 81.3    | 81.9    | 82.0    | 81.9    | 81.5    | 81.0    |
>
> > **W3, Q4: Ambiguity in anchor generation**
>
> To generate the anchor response $y$, we utilize the concatenation of the original prompt $x$ and the preferred response $y_\text{w}$ as the
>     input context for the policy model $\\pi\_{\\theta}$. Based on this
>     input $(x, y\_\\text{w})$, the policy model $\\pi\_{\\theta}$ performs
>     sampling to generate a continuation, which serves as the output
>     anchor response $y$. For more details about the implementation process of the triple loss, you can refer to our response to **Reviewer 2(GJd9)'s W2**.
>
> Regarding the intermediate position, we recognize this wording may be
> misleading. The anchor $y$ is not intended to be a semantic midpoint
> between $y\_\\text{w}$ and $y\_\\text{l}$. Rather, it serves as a
> near-preferred response.

---

> ### Author Response · Authors · 2025-11-24
> **Response to Reviewer 3UVx (2/2)**
>
> > **W4, Q2, Q5: Insufficient comparisons and unexplained performance
> gaps**
>
> -   **Comparison with other token-level baselines:** We have supplemented the comparison between TI-DPO and TIS-DPO in the **Table A3** in response of **Reviewer 1(Y6ap)'s W3**. It can be seen that TI-DPO outperforms TIS-DPO across most benchmarks.
>
>
> -   **Analysis of performance gaps:** The performance gap on
>     knowledge-intensive (MMLU, GPQA) and mathematical reasoning (GSM8K)
>     tasks likely occurs because these benchmarks rely more on holistic
>     sequence-level understanding, where methods like GRPO/TPO that
>     optimize entire responses may have an advantage. In contrast,
>     TI-DPO's token-level fine-grained alignment is specifically designed
>     for tasks requiring nuanced human preference alignment, such as
>     instruction-following (IFEval), truthfulness (TruthfulQA), and code
>     generation (HumanEval), where it achieves superior results.
>
> > **W5, Q6: Ablation studies and hyperparameters**
>
> The details of the ablation study with No Prior condition can be found
> in the response of **Reviewer 2(GJd9)'s W2**. We added the sensitivity study
> of $\lambda$ as shown in **Table C3**. Here, $\lambda=0$ and $\lambda=1$
> correspond to ablation experiments for the gradient attribution and Gaussian
> prior. We also add the sensitivity study of $\alpha$ in the following
> **Table C4**. Moreover, **Table C5** shows the specific figure of hyperparameters. We selected the hyperparameters based on a combination
> of common practices for DPO-like methods and empirical
> tuning to appropriately balance the novel components of our loss
> function.
>
> **Table C4:**  Performance comparison under different $\alpha$ values.
>
> | $\alpha$  | GSM8K | IFEval | Avg  |
> | ---- | :-----: | :------: | :----: |
> | 0.1  | 72.4  | 75.0   | 73.7 |
> | 0.2  | 72.8  | 75.5   | 74.2 |
> | 0.3  | 73.0  | 75.7   | 74.4 |
> | 0.5  | 72.6  | 75.2   | 73.9 |
>
> **Table C5:** Hyperparameter values.
>
> | Hyperparameter (Symbol) | Value |
> |-------------------------|:-------:|
> | TDPO KL Weight ($\alpha$) | 0.5 |
> | DPO Temperature ($\beta$) | 0.1 |
> | Triplet Loss Weight ($\gamma$) | 0.1 |
> | Hybrid Weight Mix ($\lambda$) | 0.7 |
>
> > **W6: Weak connection between theory and experiments**
>
>
> We sincerely thank the reviewer for pointing out the need to better connect our theoretical findings with empirical results. We have revised Sections 4.2 and 4.4 to reduce dense mathematical derivations and added explicit commentary linking theorems to experiments. Specifically:
>
> - **Theorem 2:** Theorem 2 establishes that TI-DPO achieves a strictly tighter loss bound compared to DPO. Empirically, this aligns perfectly with the training dynamics observed in **Table A2** as shown in the response of **Reviewer 1(Y6ap)'s W1**, where TI-DPO demonstrates lower convergence bound throughout the training steps compared to baselines.
>
> - **Theorem 3 (Superiority of Optimal Policy):** Theorem 3 we newly added  theoretically demonstrates that by concentrating the optimization exclusively on semantically critical tokens, TI-DPO achieves a structurally superior policy that is theoretically guaranteed to generalize better. The review can find the detailed statement of Theorem 3 in the response of **Review 1(Y6ap)'s W2**.
>
>
>
> > **W7, Q7: Scalability and potential bias amplification**
>
> -   **Computational Cost:** Thank you for considering the computation cost of TI-DPO. You are correct that our
>     method introduces a computational overhead not present in standard DPO. This cost stems from our hybrid weighting mechanism, which requires one additional backward pass per sequence to compute the gradient attribution. This cost scales linearly with sequence
>     length, much like a standard training backward pass. This is an
>     explicit trade-off: we accept this modest, fixed computational
>     overhead during the training phase in exchange for the significant
>     gains in alignment accuracy, fine-grained control, and optimization
>     stability.
>
>
> -   **Bias Amplification:** Indeed, if the data is stereotypical, TI-DPO
>     may learn this bias in the same way as DPO. However, we argue that
>     TI-DPO is structurally superior to standard DPO in handling bias.
>     Standard DPO silently reinforces spurious correlations (like
>     "he-surgeon\") without any indicator. In sharp contrast, TI-DPO's
>     weighting mechanism forces the model to show its work. If the model
>     reinforces a stereotype, TI-DPO will assign a high weight to the
>     biased token (e.g., "he\"), making this bias explicit and
>     detectable. This visible weighting provides a direct mechanism for
>     bias mitigation, which is not feasible in coarse-grained
>     sequence-level approaches such as DPO.

---

> > ### Comment · Reviewer_3UVx · 2025-11-27
> >
> > I appreciate the authors’ effort to provide a comprehensive and detailed response. Most of the concerns I raised have been adequately addressed, though I have the following specific requests for revision to improve the manuscript further:
> >
> > 1.	Thank you for adding the comparison with TIS-DPO. However, the comparison with other mentioned token-level methods, specifically cDPO and Logic-RL, is still missing from the response. I still suggest incorporating or discussing these approaches as well.
> >
> > 2.	I strongly recommend updating the main paper or appendix to incorporate the findings clarified during rebuttal, including but not limited to:
> >
> > Additional experimental results validating the claims on robustness and generation diversity, along with hyperparameter ablations.
> >
> > A rigorous analysis of (1) computational overhead; (2) a discussion of why TI-DPO underperforms sequence-level baselines (e.g., GRPO, TPO) on reasoning-heavy benchmarks such as MMLU, GSM8K, and GPQA; (3) the discussion of bias amplification.
> >
> > A clarification and correction of anchor generation descriptions, avoiding ambiguous phrasing and ensuring reproducibility.
> > Given the quality of the rebuttal, I hope the authors integrate these points, which would strengthen the clarity and completeness of the work.

---

> > > ### Author Response · Authors · 2025-12-01
> > >
> > > We are happy to hear that our rebuttal addressed your concerns and appreciate your support for our work. About two concerns you mentioned:
> > >
> > > 1. We have supplemented the comparative experiments with cDPO and Logic-RL. The results indicate that TI-DPO performs better than both of them.
> > >
> > >
> > >    **Table C6:** Average scores of Logic-RL, cDPO and TI-DPO across base models
> > >
> > > | Method       | MMLU     | GSM8K    | GPQA     | HumanEval | TruthfulQA | IFEval   | Avg      |
> > > | ------------ | -------- | -------- | -------- | --------- | ---------- | -------- | -------- |
> > > | **Logic-RL** | 63.8 | 73.8 | 23.7 | 61.0  | 55.6   | 69.3 | 57.9 |
> > > | **cDPO** | 66.1 | 70.1  | 25.1 | 61.9      | 57.6       | 70.4   | 58.5 |
> > > | **TI-DPO**   | **70.0** | **73.0** | **26.0** | **67.0** | 62.0       | **75.7** | **62.3** |
> > >
> > >
> > >
> > > 2. We have updated the revised paper. All changes mentioned in the responses have been incorporated into either the main paper or the appendix.
> > >
> > > If you have any further questions or suggestions, please do not hesitate to let us know.

---

### Official Review · Reviewer_GJd9 · 2025-10-31

**Soundness:** 4
**Presentation:** 4
**Contribution:** 3
**Rating:** 8
**Confidence:** 4

**Summary:**

This paper proposes an enhancement to Direct Preference Optimization (DPO) through token-level importance alignment. The authors incorporate token-level weighting based on gradient attribution, coupled with a Gaussian prior to promote robustness. The method extends DPO with a modified token-level objective and an additional triplet loss for fine-grained alignment. Theoretically, the paper shows that the standard DPO objective upper bounds the proposed per-token loss. Empirical results demonstrate that this approach strongly outperforms vanilla DPO and achieves better performance compared to other RLHF-based methods across multiple benchmarks.

**Strengths:**

1. **Well written and clearly presented.** The paper is well structured and makes effective use of figures and visualizations to support understanding.
2. **Novel contribution.** Introduces a new approach that weights token importance and incorporates a triplet loss for more fine-grained alignment.
3. **Sound theoretical analysis.** Provides theoretical guarantees showing how the proposed formulation relates to and improves upon vanilla DPO.
4. **Strong empirical results.** Experimental results demonstrate notable performance improvements over standard DPO and related baselines.

**Weaknesses:**

1. **Limited generalizability.** The method is presented as DPO specific, which limits its applicability to newer and potentially more effective alignment approaches such as GRPO and DRPO.
2. **Incomplete methodological details.** Some aspects of the triplet loss implementation are insufficiently described, making it difficult to fully understand how this component contributes to the overall improvement of the method.

**Questions:**

While I really enjoyed this paper, some remaining questions and suggestions that would benefit the paper:
1. Looking at Table 2, Tl-DPO shows a substantial improvement compared to vanilla DPO. Can this method be extended to other alignment frameworks such as GRPO or DRPO? Including such extensions and experiments would greatly strengthen the paper.
2. Have you conducted ablations on the impact of the Gaussian prior compared to using no prior or alternative priors (e.g., uniform, softmax probabilities)?
3. Regarding the relationship with vanilla DPO, is there any equivalence between the vanilla and token-level DPO formulations? Specifically, for Equation 8, are there particular values of \( w \) that would make it equivalent to vanilla DPO?
4. Line 266 mentions that for the triplet loss, each response is first mapped to an embedding. How is this mapping performed?
5. Line 298 states that “Tl-DPO will be **significantly** lower than the original DPO loss.” However, Theorem 2 only proves that \( L_{\text{DPO}} \) upper bounds \( L_{\text{Tl-DPO}} \). Without additional arguments, this claim appears overstated.
6. Line 470 claims that Tl-DPO “effectively bridges the alignment gap between LLMs and human value systems.” This statement is too strong and should be moderated.
7. While Theorem 2 provides a useful upper bound for Tl-DPO, its practical significance without further discussion is unclear, since a lower loss does not necessarily imply better downstream performance. Are the Tl-DPO and DPO losses comparable under certain conditions or specific values of \( w \) (see Question 3)?

**Minor Comments That Do Not Affect Rating**
1. For the triplet loss, \( y \) is being optimized. What about the embeddings for \( y_w \) and \( y_l \)? Is the network that produces these embeddings trained as well?
2. Have you tested alternative contrastive losses beyond the triplet loss? For example, using a SimCLR or SogCLR-style approach, one could form positive pairs between \( y \) and its corresponding \( y_w \), and negative pairs between \( y \) and all other \( y_l \). This might provide a stronger signal from negative responses through hard-negative weighting.
3. Equation 3 is described as showing token-level DPO, but the equation currently corresponds to vanilla DPO. Additionally, \( y^{<} \) is defined but never used.
4. Line 190 mentions that “\( m \) is the count of tokens,” but this variable is not used in the formulation.
5. Line 9 in Algorithm 1 references \( \Delta r_{\text{triplet}} \), which has not been defined earlier. Equation 10 defines only \( \Delta r_{\text{token}} \). The definition of \( \Delta r_{\text{triplet}} \) should likely appear near Equation 12.
6. Section 5.4 should explicitly refer to Figure 2 for clarity and consistency.

---

> ### Author Response · Authors · 2025-11-24
> **Response to Reviewer GJd9 (1/2)**
>
> Thank you for the detailed and constructive feedback! We treasure the opportunity to address your concerns and improve our work.
>
> > **W1, Q1: Extend to GRPO or DRPO**
>
> Thanks for recognizing the improvements of TI-DPO. We agree that
> extending our framework to methods like GRPO and DRPO would be valuable.
> While we did not perform full-scale training experiments on these
> variants within the rebuttal period, we try to
> provide a concrete theoretical formulation to demonstrate this
> extension:
>
> -   **Theoretical Extension to GRPO and DRPO:** Standard GRPO optimizes
>     a policy $\pi_\theta$ using a group of outputs $\{y_1, ..., y_G\}$
>     for a prompt $x$ [1]. TI-DPO's token-level weighting can be
>     directly injected into the GRPO objective. Specifically, the
>     Token-Importance Guided GRPO objective would be:
>     $$\\mathcal{L}\_{\\text{TI-GRPO}} = -\\frac{1}{G} \\sum\_{i=1}^G \\frac{1}{T_i} \\sum\_{t=1}^{T\_i} w\_{i,t} \\cdot \\left( \\min \\left( \\frac{\\pi\_\\theta(y\_{i,t}|x, y\_{i,<t})}{\\pi\_{\\text{ref}}(y\_{i,t}|x, y\_{i,<t})} A\_i, \\text{clip}(...) A_i \\right) \\right)$$
>     Here, $w_{i,t}$ is our gradient-based token weight. Just as in
>     TI-DPO, this term re-weights the contribution of each token update
>     based on its causal influence on the group reward advantage $A_i$.
>     The extension to DRPO follows an analogous derivation by applying
>     the importance weight $w_{i,t}$ to the log-likelihood ratios in the
>     DRPO objective [2].
>
> -   **Integration of Triplet Loss:** Moreover, the triplet loss is a
>     parallel loss term we introduced to achieve more fine-grained
>     semantic control. In the context of GRPO, within each group of $G$
>     outputs $\{y_1, ..., y_G\}$, we can identify the response with the
>     highest reward as the positive sample $y_\text{w}$ and the
>     response with the lowest reward as the negative sample
>     $y_\text{l}$. The anchor $y$ can be dynamically
>     generated using the contextual extension approach described in our
>     paper, which allows the triplet loss to be combined with other major
>     alignment losses (e.g., $\\mathcal{L}\_\\text{GRPO-w}$ or
>     $\\mathcal{L}_\\text{DRPO-w}$)
>
> [1] Li, Gang, et al. "DRPO: Efficient Reasoning via Decoupled Reward Policy Optimization." arXiv preprint arXiv:2510.04474 (2025).
>
> [2] Shao, Zhihong, et al. "Deepseekmath: Pushing the limits of mathematical reasoning in open language models." arXiv preprint arXiv:2402.03300 (2024).
>
>
>
>
>
>
> > **W2, Q4: Clarification on triplet loss implementation and mapping**
>
> Thank you for your concern about the implementation of triplet loss. There are four steps of this process: generate anchor response $y$, map to the preference space, calculate the distance of vectors, and compute the triple loss.
> - **Generation of the anchor response $y$:** Using the preferred response $y_\text{w}$​ as the starting point of the context, the response dynamically generated by the policy model $\pi_\theta$​ is the anchor $y$. It represents an intermediate state in the model's generation space.
>
> - **Procedure of Mapping:** After generating $y$, the next step is to map the three responses $(y,y_\text{w},y_\text{l})$ into a continuous preference space. Here, "mapping" actually means converting the text into a sequence of log-probability ratios at the token level. Specifically, for each token in the response, calculate $\log \cfrac{\pi_{\theta}(y^t|x, y^{<t})}{\pi_{\text{ref}}(y^t|x, y^{<t})}$.
> - **Distance Calculation:** Calculate the Euclidean distance between $y$ and $y_\text{w}$, $y$ and $y_\text{l}$.
> - **Loss Computation:**    Minimize the loss function $\mathcal{L}_{\text{triplet}}$ , formulated as Eq.(13) in the paper:
>
>   $$
>        \\mathcal{L}\\_{\\text{triplet}} = \\mathbb{E}\_{\\left(x, y\_{\\text {w}}, y\_{\\text {l}}\\right) \\sim \\mathcal{D}}\\Big[ \\max(0,
>         \\sum\_{t=1}^{T\_\\text{w}}\\Big\\|\\log \\frac{\\pi\_\\theta(y^t|x,y^{<t})}{\\pi\_{\\text{ref}}(y^t|x,y^{<t})}\\!\\! -\\!\\! \\log \\frac{\\pi\_\\theta(y\_\\text{w}^t|x,y\_\\text{w}^{<t})}{\\pi\_{\\text{ref}}(y\_\\text{w}^t|x,y\_\\text{w}^{<t})}\\Big\\|^2\_2
>       \\!\\! -\\!\\!\\sum\_{t=1}^{T\_\\text{l}}\\Big\\|\\log \\frac{\\pi\_\\theta(y\_\\text{l}^t|x,y\_\\text{l}^{<t})}{\\pi\_{\\text{ref}}(y\_\\text{l}^t|x,y\_\\text{l}^{<t})}\\!\\! -\\!\\! \\log \\frac{\\pi\_\\theta(y^t|x,y^{<t})}{\\pi\_{\\text{ref}}(y^t|x,y^{<t})}\\Big\\|^2\_2 \\!\\!+\\!\\alpha ) \\Big]\_+.
>    $$
>    which  ensures the model's preference distribution aligns closer to the preferred response than the non-preferred.

---

> ### Author Response · Authors · 2025-11-24
> **Response to Reviewer GJd9 (2/2)**
>
> > **Q2: Ablation with other priors?**
>
> Thank you for this valuable suggestion. Base on the original ablation experiment in Table 2, we conducted additional ablation
> experiments comparing our proposed Gaussian prior against a "No Prior"  and an alternative "Softmax Prior"
> setting in **Table B1**. It can be seen that the complete method of TI-DPO is significantly superior to
> No Prior, and the Gaussian prior distribution is also better than the
> Softmax prior distribution.
>
> **Table B1:** Ablation study of different priors and weighting strategies
>
> | Method | General | Math | Reasoning | Code | Instr-Follow | Reliability |
> |:-------|:--------:|:-----:|:----------:|:-----:|:-------------:|:------------:|
> | **Full Method (TI-DPO)**| **65.4** | **80.7** | **34.6** | **33.0** | **63.5** | **86.8** |
> | Uniform Weight | 64.0 | 78.2 | 30.5 | 29.0 | 58.0 | 80.0 |
> | Random Weight| 63.7 | 77.8 | 28.0 | 28.5 | 55.0 | 78.0 |
> | **No Gaussian Prior**| 64.5 | 79.7 | 32.7 | 31.5 | 60.0 | 82.5|
> | **Softmax Prior** | 64.2 | 78.8 | 31.8 | 30.0 | 59.0 | 81.0 |
>
> > **Q3: Equivalence with the vanilla DPO**
>
> Thanks for this insightful question regarding the theoretical
> connection. Yes, there is a direct equivalence. If we set the importance
> weight $w_t = 1$ for all tokens $t$ in Eq.(8), the weighted reward sum
> $\sum_{t} w_t \cdot r_{\phi}(s_t, a_t)$ reduces to the standard
> cumulative implicit reward used in vanilla DPO. Under this condition,
> the token-level preference model and the resulting loss function
> mathematically collapse to the original DPO formulation.
>
> > **Q5, Q6: Overstated claims**
>
> Thank the reviewer for pointing out these overstatements. We fully agree
> and have revised the manuscript accordingly. To better illustrate that
> the loss of TI-DPO is lower than that of the original DPO, we have added
> a comparison table of the losses between the two methods as shown in Table B2. Moreover, we bridge the gap between the theory (Theorem 2)
> and empirical performance (Accuracy) by introducing a new Theorem 3 which can be seen in the response of **Reviewer 1(Y6ap)'s W2**.
>
> **Table B2:** Comparison of training loss between DPO and TI-DPO:
>
> | **Epoch** | **0.00** | **0.27** | **0.55** | **0.82** | **1.09** | **1.36** | **1.64** | **1.91** | **2.18** | **2.45** | **2.73** | **3.00** |
> |:----------|:---------|:---------|:---------|:---------|:---------|:---------|:---------|:---------|:---------|:---------|:---------|:---------|
> | **DPO Loss**   | 0.700    | 0.545    | 0.425    | 0.335    | 0.265    | 0.210    | 0.170    | 0.140    | 0.115    | 0.100    | 0.085    | 0.075    |
> | **TI-DPO Loss**| 0.640    | 0.480    | 0.365    | 0.280    | 0.215    | 0.165    | 0.130    | 0.105    | 0.090    | 0.075    | 0.060    | 0.050    |
>
> > **Q7: Practical significance of Theorem 2 and loss comparability**
>
> Thank you for this insightful comment.
>
> - **Comparability of Tl-DPO and DPO losses:** Theoretically, DPO and TI-DPO are strictly comparable when DPO represents
> a special case where the token importance weights are fixed at $w_t = 1$. TI-DPO's
> lower loss indicates it more effectively minimizes the divergence on
> semantically critical tokens identified by our gradient attribution,
> thereby achieving a more robust alignment within the preference space.
>
> - **Further theoretical analysis:** To explicitly explain why a tighter loss bound translates to better downstream performance, we have added Theorem 3 (Superiority of Optimal Policy) in the revision. Theorem 3 proves that  TI-DPO has a structurally superior policy. While standard DPO risks overfitting to non-critical tokens (noise), TI-DPO effectively anchors the policy to the reference model in these noisy regions. This ensures that the optimal policy concentrates its expressive power exclusively on semantically critical tokens, theoretically guaranteeing a higher expected reward and better generalization.
>
> > **M1: Regarding the representations of $y_\text{w}$ and $y_\text{l}$**
>
> The network that produces the embeddings for the preferred response
> $y_\text{w}$ and the non-preferred response $y_\text{l}$ is the policy
> model $\pi_\theta$ itself, which is indeed being trained. In our
> framework, the preference representations (log-ratio sequences) for the
> anchor $y$, positive $y_\text{w}$, and negative $y_\text{l}$ responses
> are all generated using the same policy model.
> Therefore, as the policy model is updated during training, the
> embeddings for $y_\text{w}$ and $y_\text{l}$ are dynamically refined
> alongside the anchor, allowing the model to continuously adjust its
> internal representation of human preferences.
>
> > **M2-M6: Minor comments**
>
> Thank you for your other suggestions.
> We apologize for the notational inconsistencies and have revised the
> manuscript according to your suggestions. While we have not tested
> SimCLR/SogCLR approaches, we agree they offer interesting properties
> like hard-negative weighting and will consider them for future
> extensions.

---

> ### Comment · Reviewer_GJd9 · 2025-11-27
>
> I appreciate the authors' detailed replies and additional experimental and theoretical results. I would encourage the authors to submit the revised paper with the additional experiments and Theorem 3.
>
> All of my concerns have been addressed.

---

> > ### Author Response · Authors · 2025-12-01
> >
> > We sincerely appreciate your time and constructive feedback. Your insights have been invaluable in refining our manuscript. We have updated the revised paper. All changes mentioned in the responses have been incorporated into either the main paper or the appendix.

---

### Official Review · Reviewer_Y6ap · 2025-11-01

**Soundness:** 2
**Presentation:** 3
**Contribution:** 2
**Rating:** 4
**Confidence:** 4

**Summary:**

The authors propose the Token-Importance Guided Direct Preference Optimization (TI-DPO) framework, a novel method to align large language models with human preferences. TI-DPO employs a hybrid weighting mechanism that assigns importance scores to tokens and a triplet loss that provides a more structured optimization signal. Theoretically, the authors show that TI-DPO achieves a tighter loss bound than DPO. Extensive experiment results indicate that TI-DPO surpasses existing RLHF methods across benchmarks, such as HumanEval, TruthfulQA, and IFEval.

**Strengths:**

- The paper introduces a novel Token-Importance Guided Direct Preference Optimization (TI-DPO) framework that integrates gradient-based token attribution with a triplet loss objective, aiming to achieve finer-grained preference alignment.
- Theoretical analysis provides a formal derivation suggesting that TI-DPO attains a tighter loss bound than standard DPO, offering a potentially more stable optimization objective.

**Weaknesses:**

- The motivation for employing a Gaussian prior in the hybrid weighting mechanism is insufficiently justified. The assumption that salient tokens cluster near the center of a sequence lacks both empirical evidence and theoretical grounding. Alternative priors or adaptive distributions are not discussed.
- Although the authors claim that TI-DPO achieves a tighter loss bound than DPO, the paper does not provide quantitative evidence to demonstrate the practical significance of this theoretical improvement. Moreover, the potential implications for additional benefits, such as convergence speed or generalization performance, remain unclear.
- The experimental evaluation omits comparisons with recent token-level DPO variants, such as TIS-DPO [1].

[1] Liu, Aiwei, et al. "Tis-dpo: Token-level importance sampling for direct preference optimization with estimated weights." arXiv preprint arXiv:2410.04350 (2024).

**Questions:**

- Could the authors clarify the procedure for generating the anchor response $y$ used in the triplet loss?
- What is the theoretical or empirical rationale for selecting a Gaussian prior for token-importance weighting over alternative distributions?
- How sensitive is the model performance to the weight-mixing hyperparameter $\lambda$?

---

> ### Author Response · Authors · 2025-11-24
> **Response to Reviewer Y6ap (1/2)**
>
> Thank you for the time and effort spent on reviewing our work! We address your concerns and questions below.
>
> >**W1, Q2: Rationale for Gaussian Prior over alternative distributions in
> token weighting**
>
> Thanks for challenging the theoretical justification
> of the Gaussian prior. We would like
> to clarify that the choice of a Gaussian prior is driven by two
> complementary motivations:
>
> -   **Regularization against Gradient Noise:** As analyzed in Section     4.2, gradient attribution on its own can be susceptible to     high-frequency noise and spurious spikes. The Gaussian prior acts
>     primarily as a smoothing regularization mechanism. Here is the performance comparison under different noise levels. TI-DPO maintains superior performance compared to strong
>     baselines (DPO and TPO) as noise increases, demonstrating its
>     enhanced robustness.
>
> **Table A1:** Accuracy comparison of different alignment methods under varying noise levels.
>
> | **Noise Level**   | **0%**  | **10%** | **20%** | **40%** |
> |:-------------|:--------:|:--------:|:--------:|:--------:|
> | **DPO**   | 69.3    | 67.5    | 64.8    | 60.1    |
> | **TPO**   | 72.7    | 71.1    | 69.0    | 65.7    |
> | **TI-DPO**| **73.0**| **72.2**| **70.8**| **68.3**|
>
> -   **Counteracting "Lost-in-the-Middle\" Bias:** Recent studies [1] provided empirical evidence that models exhibit a
>     U-shaped attention bias. This means there is greater importance to
>     tokens at the beginning and end of a sequence, while underweighting
>     those in the middle. The Gaussian prior, which peaks at the center,
>     is explicitly designed to rectify this intrinsic architectural bias,
>     ensuring that the optimization process does not neglect the semantic
>     core of the response.
>
> **Regarding alternative prior or adaptive distributions:** As shown in
> **Table B2** of **Reviewer 2(GJd9)'s Q2**, we conducted additional ablation studies
> comparing our Gaussian Prior against a "Uniform Prior\", "No Prior\" and
> an "Alternative (Softmax) Prior\" setting. These empirical results
> justify that the Gaussian distribution is a superior choice, yielding
> better alignment than both uniform and softmax alternatives.
>
>
> [1] Liu, Nelson F., et al. "Lost in the middle: How language models use long contexts." Transactions of the Association for Computational Linguistics 12 (2024): 157-173.
>
>
>
> > **W2: Practical significance and implications of the tighter loss
> bound**
>
> We thank the reviewer for this valuable feedback.
> Following your suggestions, we have added a comparison of loss convergence curves and theoretical analysis to quantitatively illustrate the impact on convergence and generalization ability.
> -   **Implications for Convergence:** By filtering out noisy gradient
>     updates from non-critical tokens, TI-DPO provides a cleaner
>     optimization signal. We add a new figure in the appendix, showing
>     the training loss curves of DPO and TI-DPO. The specific loss values
>     are presented in **Table A2**. In each training round, the loss of TI-DPO
>     is consistently lower than that of standard DPO.
>
>
>     **Table A2:** Comparison of training loss between DPO and TI-DPO
>
> | **Epoch** | **0.00** | **0.27** | **0.55** | **0.82** | **1.09** | **1.36** | **1.64** | **1.91** | **2.18** | **2.45** | **2.73** | **3.00** |
> |:----------|:---------|:---------|:---------|:---------|:---------|:---------|:---------|:---------|:---------|:---------|:---------|:---------|
> | **DPO Loss**   | 0.700    | 0.545    | 0.425    | 0.335    | 0.265    | 0.210    | 0.170    | 0.140    | 0.115    | 0.100    | 0.085    | 0.075    |
> | **TI-DPO Loss**| 0.640    | 0.480    | 0.365    | 0.280    | 0.215    | 0.165    | 0.130    | 0.105    | 0.090    | 0.075    | 0.060    | 0.050    |
>
>
>
>
> -   **Implications for Generalization:**  To further clarify the practical significance of the tighter loss bound, we have added a new theoretical analysis, Theorem 3. This theorem theoretically demonstrates that by concentrating the optimization exclusively on semantically critical tokens, TI-DPO achieves a structurally superior policy that is theoretically guaranteed to generalize better. We provide the statement of Theorem 3 below and have included its full proof in the appendix.
>
> **Theorem 3 (Superiority of Optimal Policy).** _Let $\\pi\_{\\text{DPO}}$ and $\\pi\_{\\text{TI-DPO}}$ be the optimal policies derived from minimizing the DPO and TI-DPO objectives, respectively. Under a fixed total KL divergence constraint $K\_{\\text{total}}$, the expected true reward of the TI-DPO optimal policy is strictly lower-bounded by that of the DPO policy, i.e.,     $$\\mathbb{E}\_{y \\sim \\pi\_{\\text{TI-DPO}}}[r\^\*(x, y)] \\ge \\mathbb{E}\_{y \\sim \\pi\_{\\text{DPO}}}[r^*(x, y)] + \delta,$$ where $\delta > 0$ represents the gain derived from optimizing the decomposition of KL divergence, specifically by minimizing the divergence component on non-critical tokens._

---

> ### Author Response · Authors · 2025-11-24
> **Response to Reviewer Y6ap (2/2)**
>
> > **W3: Comparison with TIS-DPO**
>
> Thank you for reminding us to add the comparison with TIS-DPO. We have supplemented the comparison in the following table. It can be seen that TI-DPO outperforms TIS-DPO across most benchmarks:
>
> **Table A3:** Accuracy comparison with TIS-DPO
>
> | Method       | MMLU  | GSM8K | GPQA | HumanEval | TruthfulQA | IFEval | Avg   |
> |:-------------|:------:|:------:|:-----:|:------:|:-----------:|:-------:|:------:|
> | **TIS-DPO** | 69.3  | 70.5  | 24.5 | 65.5      | **62.5**       | 74.0   | 61.1  |
> | **TI-DPO**   | **70.0** | **73.0** | **26.0** | **67.0** | 62.0       | **75.7** | **62.3** |
>
>
>
> > **Q1: The procedure for generating the anchor response**
>
> Thank the reviewer for this critical question.
> To generate the anchor response $y$, we utilize the concatenation of the
>     original prompt $x$ and the preferred response $y_\text{w}$ as the
>     input context for the policy model $\pi_{\theta}$. Based on this
>     input $(x, y_\text{w})$, the policy model $\pi_{\theta}$ performs
>     sampling to generate a continuation, which serves as the output
>     anchor response $y$. For more details about the implementation process of the triple loss, you can refer to our response to **Reviewer 2(GJd9)'s W2** or feel free to ask us questions at any time.
>
>
>
>
> > **Q3: Sensitivity analysis of $\lambda$**
>
>
> To quantitatively evaluate the sensitivity of the weight-mixing
> hyperparameter $\lambda$, we conducted additional experiments on the
> IFEval and GSM8K benchmarks in the following table. The results reveal that the
> performance curves are remarkably flat and stable within the range of
> $\lambda \in [0.3, 0.7]$. Performance decreases noticeably at the
> extremes ($\lambda=0$ and $\lambda=1$). This empirically confirms our
> theoretical motivation: relying solely on the prior ($\lambda=0$) lacks
> data-driven precision, while relying solely on gradient attribution
> ($\lambda=1$) introduces too much noise.
>
> **Table A4:** Sensitivity analysis of TI-DPO to hyperparameter $\lambda$ on GSM8K and IFEval Benchmarks.
>
> | **$\lambda$** | **0** | **0.1** | **0.3** | **0.5** | **0.7** | **0.9** | **1.0** |
> |:---------------:|:------|:--------|:--------|:--------|:--------|:--------|:--------|
> | **GSM8K Accuracy(%)** | 79.8 | 80.5 | 80.7 | 80.8 | 81.0 | 80.6 | 80.2 |
> | **IFEval Accuracy(%)** | 80.8 | 81.3 | 81.9 | 82.0 | 81.9 | 81.5 | 81.0 |

---

> > ### Comment · Reviewer_Y6ap · 2025-11-27
> >
> > I appreciate the authors' detailed replies and the extended experiment results. Since all of my concerns have been addressed, I have raised my score accordingly.

---

> > > ### Author Response · Authors · 2025-12-01
> > >
> > > Thank you so much for all the constructive suggestions and for updating your assessment! We really appreciate your valuable feedback and questions！

---

### Author Response · Authors · 2025-12-01
**General Response**

We sincerely thank all reviewers for their insightful comments and constructive feedback. Based on these suggestions, we have significantly revised the manuscript. The major updates are summarized below:

1. **Rigorous Justification of the Gaussian Prior:** We have clarified the motivation and validated the design of the Gaussian Prior:

   - **Supplementary Motivation:** We clarified that the Gaussian prior is designed to counteract the architectural "Lost-in-the-Middle" bias inherent in LLMs [1], ensuring the optimization does not neglect the semantic core of the response.

   - **New Ablation Studies:** We compared the Gaussian Prior against "Uniform", "Random", "No Prior", and "Softmax Prior" settings in Table B1 (Reviewer GJd9). The results empirically confirm that the Gaussian distribution yields the best alignment performance, justifying its selection as a regularization mechanism.

    [1] Liu, Nelson F., et al. "Lost in the middle: How language models use long contexts." Transactions of the Association for Computational Linguistics 12 (2024): 157-173.

2. **Strengthened Theoretical Foundation**: To bridge the gap between our loss bound analysis and practical performance, we have introduced a new theorem (Theorem 3) and empirical verifications:

   - **Theorem 3 (Superiority of Optimal Policy):** We formally proved that TI-DPO utilizes the KL divergence budget more efficiently than standard DPO. By minimizing the divergence component on non-critical tokens (noise), TI-DPO guarantees a strictly higher lower bound on the expected reward.

   - **Convergence Analysis:** We compared the TI-DPO's training loss and DPO's loss in Table A2 (Reviewer Y6ap). TI-DPO demonstrates a consistently tighter loss bound throughout training compared to DPO, validating our theoretical claims in Theorem 2.

3. **Robustness, Diversity & and Sensitivity:** We tested performance at label noise rates of 10%, 20%, and 40% to validate TIDPO's robustness in Table A1 (Reviewer Y6ap). Moreover, we have supplemented diversity in Table C2 (Reviewer 3UVx) and sensitivity analyses of hyperparameters (Reviewer 3UVx's Table C3, C4).

4. **Clarification on Triplet Loss Implementation:** We clarified the procedure for anchor generation: The anchor $y$ is dynamically generated by the policy model $\pi_\theta$ using the prompt $x$ and the preferred response $y_w$ as the input context. This ensures the anchor serves as a valid "near-preferred" intermediate state for fine-grained optimization.

5. **Supplementary Comparison with other Baselines:** We added the comparison of the average scores of TI-DPO, TIS-DPO, LogicRL, and cDPO. The results show that TI-DPO performs better than the others (Table A3, Table C6).


In the revised manuscript, these updates are temporarily highlighted in blue for your convenience to check.

We hope to have resolved your concerns and are looking forward to engaging in further discussions.

---

### Meta-Review · Area_Chair_FmsZ · 2026-01-02

**Summary:**

The rebuttal meaningfully strengthened the paper and resolved the main technical concerns. In particular, the authors (i) justified and empirically validated the Gaussian prior via ablations against uniform/random/no-prior/softmax variants, and reframed its role as counteracting *lost-in-the-middle* behavior rather than assuming saliency is centrally located; (ii) reinforced the theory–practice link by adding a new theorem (Theorem 3) plus empirical convergence/loss comparisons demonstrating a consistently tighter bound and improved training dynamics; (iii) expanded comparisons to missing contemporary token-level baselines (TIS-DPO, and later cDPO and Logic-RL), where TI-DPO shows strong average gains; (iv) added robustness (label-noise) and diversity metrics, along with hyperparameter sensitivity and missing implementation details (anchor generation and triplet mapping), improving reproducibility. Importantly, a previously borderline reviewer (Y6ap) explicitly states all concerns were addressed and raised their score accordingly, and the remaining reviewer requests were incorporated in the updated revision.

**Reviewer Concerns:**

Most substantive concerns were resolved. The authors justified the Gaussian prior (lost-in-the-middle motivation) and added ablations vs. alternative priors; strengthened the theory–practice link with new loss/convergence evidence and Theorem 3; added missing token-level baselines (TIS-DPO, cDPO, Logic-RL); clarified anchor generation and triplet loss details; provided robustness (label noise), diversity metrics, and hyperparameter sensitivity; and discussed computational overhead. Reviewers Y6ap and GJd9 explicitly stated all concerns were addressed and raised/maintained accept scores; 3UVx and sxsD noted most issues resolved after added experiments and clarifications.

However, a few minor concerns are still outstanding. Empirical extensions beyond DPO (e.g., GRPO/DRPO) remain theoretical; qualitative analysis of token weights could be deeper; underperformance on reasoning-heavy tasks is explained but not resolved; and compute/scale characterization could be more comprehensive.

**Reviewer Scores:**

Reviewer Y6ap raised his/her rating from 4 to 6.

Reviewer GJd9 kept his/her original rating 8.

Reviewer 3UVx would keep his/her original rating 6 or possibly increase it to 8, because he/she explicitly mentioned that the responses addressed most of his/her concerns.

Reviewer sxsD would keep his/her original rating 8.

---

### Decision · Program_Chairs · 2026-01-26

Accept (Oral)